# Wide-swath satellite altimetry unveils global submesoscale ocean dynamics

Matthew Archer[1✉], Jinbo Wang[1,2✉], Patrice Klein[1,3,4], Gerald Dibarboure[5] & Lee-Lueng Fu[1]

Ocean submesoscale (1–100 km) processes and their substantial impact on Earth's climate system have been increasingly emphasized in recent decades by high-resolution numerical models and regional observations[1–11]. However, the dynamics and energy associated with these processes, including submesoscale eddies and nonlinear internal waves, have never been observed from a global perspective. Where, when and how much do these submesoscale processes contribute to the large-scale ocean circulation and climate system? Here we show data from the recently launched Surface Water and Ocean Topography (SWOT) satellite[12] that not only confirm the characteristics of submesoscale eddies and waves but also suggest that their potential impacts on ocean energetics, the marine ecosystem, atmospheric weather and Earth's climate system are much larger than anticipated. SWOT ushers in a new era of global ocean observing, placing submesoscale ocean dynamics as a critical element of the Earth's climate system.

Ocean circulation, governed by geophysical flow dynamics, is linked to dynamic sea surface height (SSH) that can be measured by radar altimeters from space. Since the Seasat mission launched in 1978, satellite altimetry has been used to study ocean dynamics at scales greater than 100 km, including the large-scale ocean circulation, mesoscale eddies[13] and linear internal tidal waves[14]. Mesoscale eddies, generated through the release of potential and kinetic energy from larger scales, account for more than 80% of the ocean's kinetic energy[15]. These eddies act as conduits for the energy input at large scales, by solar heating and wind, to cascade towards the smallest scales at which dissipation takes place[1]. They have a profound influence on the ocean's energy balance[1], atmospheric circulation[16–18] and marine ecosystems[19,20]. Internal tidal waves are generated by the interaction between astronomical tides and bathymetry in a stratified ocean. These waves transport energy over long distances as propagating perturbations of isopycnals (constant density surfaces) to eventually dissipate through mixing and turbulence in remote locations[8,9].

However, conventional satellite altimetry is unable to resolve scales smaller than 100 km in two dimensions owing to instrument noise and one-dimensional sampling. The lack of global observations in the 1–100-km scale range remains a limiting factor to understand and quantify the role of the ocean in the global climate system and to accurately predict future changes[2,3,21,22].

The presence of submesoscale ocean turbulence and nonlinear waves has long been evident in satellite imagery. Notable examples include the sunglint images of eddies captured by Apollo mission astronauts, famously described as 'spirals on the sea' by Munk[23], and the sharp, elongated internal solitary waves observed in Seasat synthetic aperture radar (SAR) images[24]. In contrast to their well-documented spatial structures, the dynamic properties of these features, such as velocity (kinetic energy) and pressure (potential energy) fields, have never

been widely observed. Yet, state-of-the-art numerical models and regional field observations indicate that they greatly influence ocean energetics. Internal solitary waves, a nonlinear class of internal tidal waves, are characterized by large amplitudes and propagate energy over long distances before dissipating, leading to mixing at smaller scales[9]. Submesoscale currents, although less energetic than their mesoscale counterparts, play a pivotal role in both the inverse kinetic energy cascade and forward energy dissipation towards microscales[3,25]. Furthermore, because of their strong horizontal gradients, they contribute substantially to vertical transport[5,26,27]; "Just as thin ducts in the lung called alveoli facilitate the rapid exchange of gases when breathing, fronts are the ducts through which heat, carbon, oxygen, and other climatically important gases enter into the deep ocean"[22]. Submesoscale eddies also support and sustain marine ecosystems by driving cold, nutrient-rich waters from depth upward to the surface, where they can be used by phytoplankton and zooplankton[4,27,28].

But to what extent are global numerical simulations accurate and how representative are regional observations on a global scale? Recent studies have shown alarming changes to the mean state and variability of the ocean, yet the implications remain uncertain[29,30]. Our ability to predict future change is closely linked to our capability to observe the ocean dynamics in sufficient detail[1].

## Wide-swath satellite altimetry

Here we present the first global measurements of the dynamic ocean at the submesoscale, from the highly anticipated SWOT wide-swath altimetry mission, launched on 16 December 2022 after 20 years of development. This pioneering satellite was designed to capture ocean dynamics down to 15-km wavelength through measuring global SSH (equivalent pressure)[12] but surprised the community with first-light SSH

[1]Jet Propulsion Laboratory, California Institute of Technology, Pasadena, CA, USA. [2]Texas A&M University, College Station, TX, USA. [3]Department of Environmental Science and Engineering, California Institute of Technology, Pasadena, CA, USA. [4]LMD-IPSL, ENS, PSL Université, Ecole Polytechnique, Sorbonne Université, CNRS, Paris, France. [5]Centre National d'Études Spatiales, Toulouse, France. ✉e-mail: archer@jpl.nasa.gov; jinbo.wang@tamu.edu

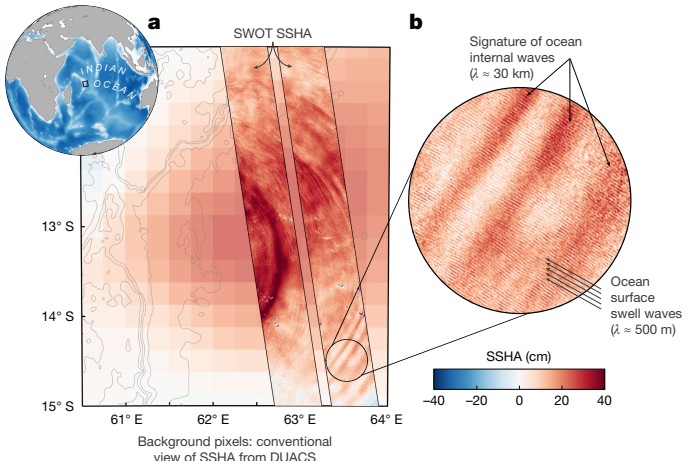

**Fig. 1 | The advance of SWOT beyond conventional altimetry.** An example in the east Indian Ocean at the Mascarene Plateau on 8 May 2023. **a**, Globe inset shows ocean bathymetry from ETOPO1 data; the black box denotes the Mascarene Plateau region. The main panel shows a background map of DUACS SSHA generated by AVISO+ using conventional altimetry data, superimposed with a swath from SWOT L2 unsmoothed SSHA. DUACS provides mapped spatial coverage at low resolution (about 150 km in space and about 15 days in time). SWOT provides a swath, limited in space, at high resolution (about 1 km in space and near-instantaneous in time). **b**, Zoom-in of the swath reveals the very high resolution of SWOT, beyond conventional altimetry: nonlinear internal waves and surface swell waves are captured in the SWOT SSHA measurements.

images that returned with great clarity beyond expectations (Fig. 1), achieving a spatial resolution of $O$(1 km). This is made possible by the Ka-band Radar Interferometer (KaRIn)[31], flown in space for the first time, which marks a notable advance in altimetry[32]. KaRIn offers an order of magnitude reduction in instrument noise compared with conventional altimeters. It also collects a two-dimensional swath of 120 km, surpassing the one-dimensional sampling capability of conventional altimeters. In this paper, we highlight SWOT's capability beyond what has been possible up to now: a global statistical view of ocean small-scale variability, and two quantitative case studies: computing the energy flux from nonlinear internal solitary waves and estimating the vorticity and vertical velocity of a submesoscale eddy.

## SSH beyond conventional altimetry

SWOT reveals previously unseen ocean processes in remarkable detail. Figure 1 shows one example from an abundance of small-scale features across the globe, near the Mascarene Plateau, an underwater topographic rise in the Indian Ocean. The existing state-of-the-art two-dimensional SSH product (Data Unification and Altimeter Combination System (DUACS) SSH, shown as background pixels in Fig. 1; see Methods) captures a mesoscale anticyclonic eddy with a positive sea surface height anomaly (SSHA) of diameter approximately 150–200 km. On the same day, instantaneous SWOT observations unveil the presence of not only the mesoscale eddy but a wealth of submesoscale features, including waves radiating eastward away from the plateau. The waves are nonlinear internal solitary wave packets with wavelengths ranging from several to tens of kilometres and SSHA amplitudes up to 30 cm. Solitary waves are known to be generated by tidal flow across a sill in the plateau[33]. Although they have been observed in SAR images, their amplitudes have never been measured from space, until SWOT.

SWOT can resolve beyond even submesoscale processes: storm-generated surface swell waves with wavelengths of several hundred metres are revealed in the data (Fig. 1). Swells were not expected to be observed because the onboard processing algorithm was designed

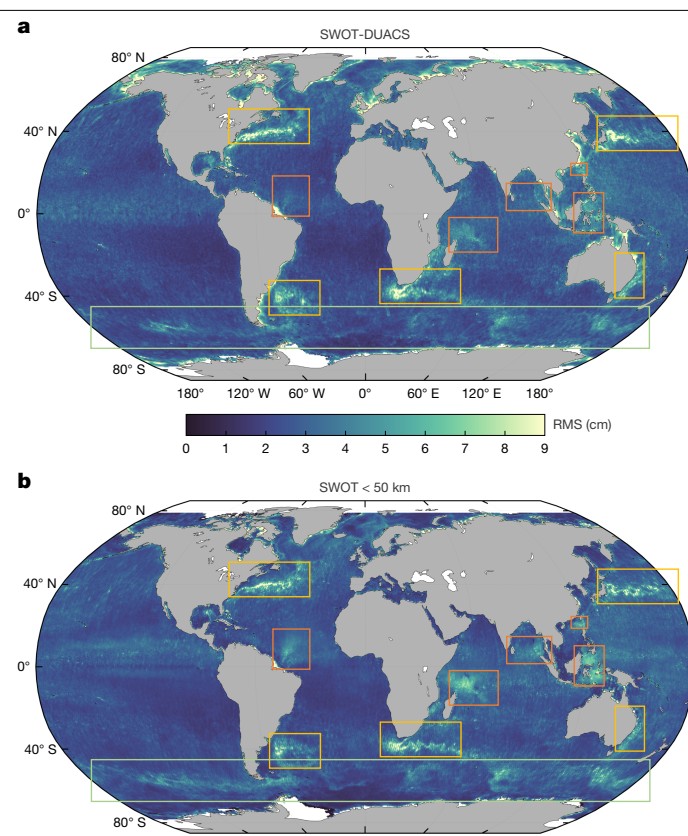

**Fig. 2 | A new global view of ocean submesoscale variability from SWOT.** Submesoscale variability as defined here includes currents and waves less than about 100 km. **a**, RMS of the difference between SWOT and DUACS SSHA over the global ocean. **b**, RMS of the 50-km high-passed SWOT SSHA over the global ocean. Orange boxes denote regions of internal tidal wave activity, yellow boxes denote western boundary current regions and the green box denotes the Antarctic Circumpolar Current, all of which are regions of elevated submesoscale SSHA variability, shown for the first time from global observations by SWOT.

to filter them out. However, owing to the high signal-to-noise ratio of KaRIn, the long swell wave signals pass through the filter, appearing with heavily suppressed amplitudes[34]. Figure 1 offers a glimpse of SWOT's unprecedented capability—its SSH measurements encompass a wide range of spatial scales from thousands of kilometres down to hundreds of metres.

A global view of submesoscale variability—captured for the first time by SWOT—is shown in Fig. 2. It is computed by means of two complementary approaches (see Methods). In the first (Fig. 2a), SWOT SSHA beyond conventional altimetry is defined as the root mean square (RMS) difference between SWOT and DUACS SSHA. Notable differences appear in high-kinetic-energy regions such as western boundary currents (Fig. 2, yellow boxes), including the Gulf Stream and Kuroshio Extension, and the Antarctic Circumpolar Current (green box). Large differences are also observed at internal gravity wave generation sites—particularly the Amazon River mouth, Mascarene Plateau, South China Sea, Andaman Sea and Indonesian archipelago (orange boxes)—as well as in tropical oceans and coastal zones. This collocation of the largest differences with ocean dynamical regimes indicates that they are related more to ocean physics than measurement noise or errors (see Methods). However, these differences can be the result of processes with short scales in space or time, especially in regions with rapidly evolving dynamics such as the tropics, wave-dominated areas and

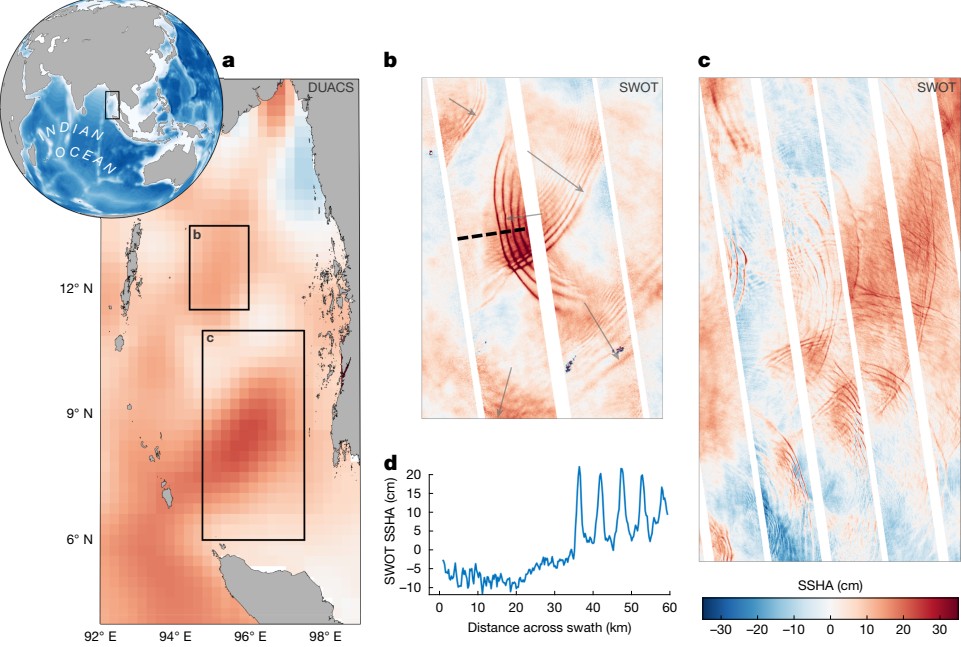

**Fig. 3 | SWOT measures the amplitudes of internal solitary waves in the Andaman Sea. a**, Globe inset shows ocean bathymetry from ETOPO1 data; the black box denotes the Andaman Sea region. The main panel shows DUACS SSHA on 1 February 2024. The two black boxes indicate regions depicted in panels **b** and **c**. **b**, Zoom-in from SWOT cycle 10 descending passes shows internal solitary waves propagating in different directions (delineated by grey arrows) that pass through one another while retaining their shape. The dashed black line indicates the transect shown in **d**. **c**, Zoom-in to show more SWOT swaths that capture the intricate wave field of the Andaman Sea. **d**, An across-swath transect line through a westward-propagating wave packet, with wave amplitudes of 20 cm and wavelengths of approximately 5 km.

western boundary currents. Time differences are a factor because the SWOT data are near-instantaneous, whereas the DUACS SSHA data are smoothed in time during mapping (between 10 and 30 days, depending on latitude)[35].

The second approach (Fig. 2b) solves the entanglement of time and space signals by isolating small spatial scales through the RMS of 50-km high-passed SWOT SSHA (see Methods). This approximates what SWOT observes beyond the conventional nadir altimeters that measure one-dimensional SSH at a resolution between 50-km and 70-km wavelength. This further emphasizes the submesoscale SSHA variability of current systems and internal wave hotspots now revealed by SWOT.

For a closer look at these new features, we present two examples: an internal solitary wave packet and a submesoscale coherent eddy. SWOT captures the SSH signatures of such features in abundance, yet they are often comingled in space; the following examples were chosen for the singular spatial dominance of waves or currents in the SSH field (see Methods).

## Internal solitary waves

Internal solitary waves are extremely clear in SWOT imagery (Figs. 1 and 3). In the Andaman Sea (Fig. 3), the solitary waves have large amplitudes of 20 cm SSHA and wavelengths of 5 km. If we were to compute the geostrophic currents directly from these observations without removing such features, it would yield a current speed of 12 m s$^{-1}$! They retain their shape and speed after colliding with one another—a special characteristic of solitary waves[36] (Fig. 3b,c). These features have been observed for many years as fine radiating wave patterns in SAR and sunglint imagery[37,38], but the fundamental question for ocean energetics can only be answered by measuring their amplitude[39]: what is the amount and variability of their energy content and fluxes? This is essential for understanding the ocean's energy budget but which was not possible to answer until SWOT. Using the SSHA measured by SWOT (Fig. 3c)

and stratification profiles from climatology, the vertically integrated time-averaged linear energy flux of these solitary waves is 8 kW m$^{-1}$ for a solitary wave packet with 20-cm peak amplitude and 1.8 kW m$^{-1}$ with for 10-cm peak amplitude (see Methods)—much larger than the value of 0.8 kW m$^{-1}$ from the coherent M$_2$ internal tides in this region[40].

## Submesoscale eddies

A submesoscale eddy observed by SWOT offshore of South Africa is completely missed in the conventional DUACS product (Fig. 4a). This is one of many such cases (for example, Extended Data Fig. 2). The eddy has an SSHA amplitude of 15 cm but a radius of only about 15 km! The eddy was also mapped by near-simultaneous high-resolution satellite infrared and colour imagery (Fig. 4b,c). It has a core 2.5 °C colder than its surrounding area, which indicates an intense lateral buoyancy gradient and nonlinearity. Velocity can be derived from SSH using the geostrophic approximation (with strong precautions; see Methods)—the predominant force balance of the ocean between the pressure gradient force and the effect of Earth's rotation—and it exceeds 1 m s$^{-1}$. However, the Rossby number—a measure of the relative importance of inertial and Coriolis forces, defined as Ro = $U/fL$ (in which $U$ is the velocity scale, $f$ is the Coriolis frequency and $L$ is the length scale)—exceeds 0.5 in the eddy core, which is a clear indication that the eddy is not in geostrophic balance (for which $R \ll 1$; see Methods). Given the small radius and large SSHA amplitude of this eddy, this is not surprising. A better approximation to compute the velocity of this eddy is cyclogeostrophy, which adds a third term to the force balance: centrifugal acceleration. This term is needed because the strong curvature of the currents around the low-pressure centre requires an acceleration to maintain balance. For a cyclonic eddy, the centrifugal force combines with the pressure gradient force, resulting in smaller angular velocities compared with geostrophic balance (for this eddy, it decreases to roughly 0.5 m s$^{-1}$). Such a high Rossby number makes this eddy coherent and persistent

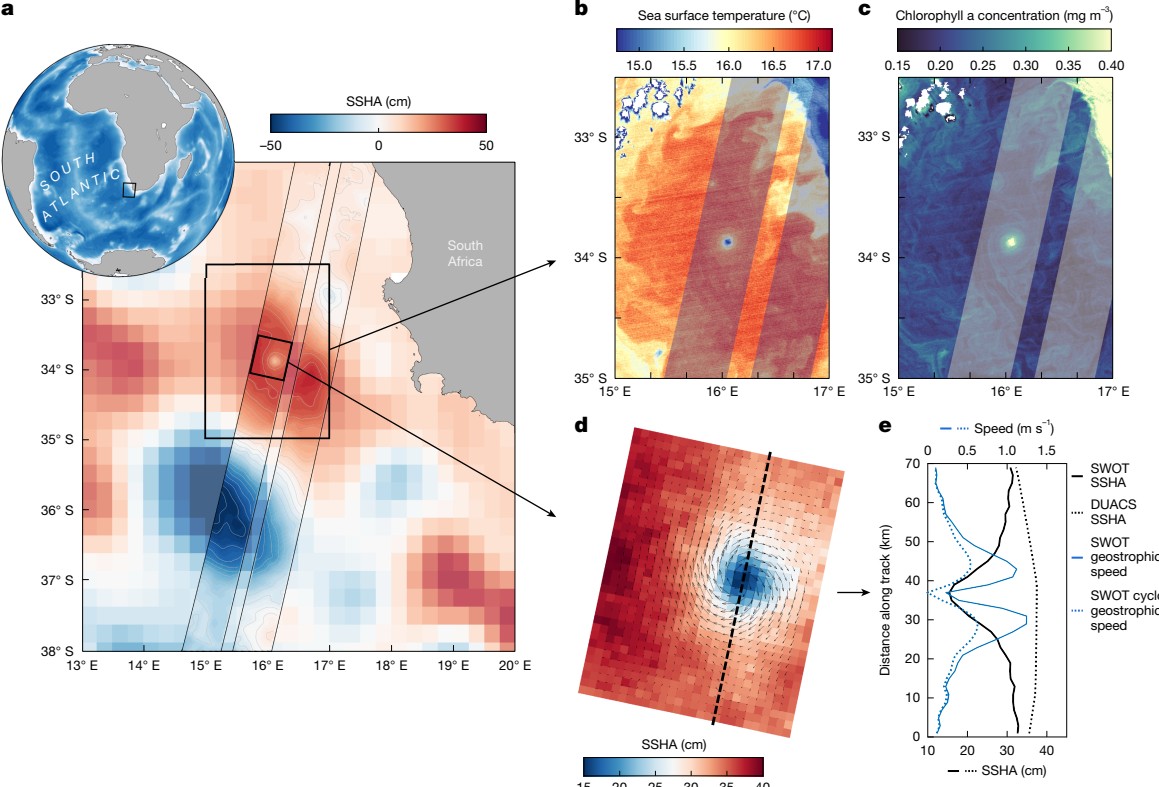

**Fig. 4 | SWOT captures a submesoscale eddy off South Africa missed in conventional altimetry. a**, Globe inset shows ocean bathymetry from ETOPO1 data; the black box denotes the study site offshore of South Africa. The main panel shows DUACS SSHA with an ascending SWOT swath superimposed, on 2 August 2023. Grey contours denote SWOT SSHA. **b**,**c**, Black boxes in **a** denote a coherent eddy with strong signatures of sea surface temperature (**b**) and chlorophyll a concentration (**c**) measured by the Suomi National Polar-orbiting Partnership (NPP) Visible Infrared Imaging Radiometer Suite (VIIRS) satellite at 750-m resolution. **d**, Zoom-in to the left SWOT swath showing SSHA and plotted on top are geostrophic velocity vectors. The dashed black line shows the along-swath transect plotted in the line plot in panel **e**. **e**, The SSHA eddy signature from SWOT and DUACS (which misses the eddy) and SWOT current speed based on geostrophy and cyclogeostrophy.

(preventing its destruction by mixing) until it encounters another eddy, which eventually leads to merging to produce a larger eddy[1].

Analysis of SWOT SSH in other high-kinetic-energy regions confirms the ubiquitous presence of turbulent submesoscale eddies, coherent and persistent, interacting with energetic mesoscale eddies. These submesoscale eddies can have diameters less than 10 km with Rossby numbers larger than 2 (see Extended Data Figs. 1 and 2, in the Kuroshio Extension). The submesoscale eddies have RMS SSHA values about three times larger than those from the highest-resolution global ocean simulations (Extended Data Fig. 3): these SWOT observations suggest that the contribution of submesoscale turbulence to the Earth's climate system is much more important than previously anticipated.

Submesoscale eddies are characterized by strong velocity gradients and intense eddy interactions that explain much of the vertical velocity field in the upper ocean (see Methods). The consequences are substantial. Upward vertical heat transport associated with submesoscale eddies is 5–10 times larger than for mesoscale eddies[5]. This warms the ocean surface and increases the moisture supply to the atmosphere[41]. We can theoretically use SWOT observations to derive vertical velocities at a given depth from the surface divergence, assuming that this divergence is constant down to this depth (see Methods): we estimate the vertical velocity for an eddy in the Kuroshio Extension that was observed by SWOT twice in 12 h (Extended Data Fig. 4) to range between −6 and −14 m per day. However, on the basis of the very large magnitudes of submesoscale SSHA gradients in this region (Extended Data Figs. 2 and 3), we expect that values can be much greater than this example at other times, as found in regional observations[26,42–44]. The first SWOT results confirm the impact of submesoscales on ocean dynamics and heat budget, based on SSHA gradient values much larger than those found from numerical models.

## Opportunities and challenges of SWOT

SWOT's precise SSH measurements of ocean submesoscale processes enable us to quantify their dynamics for the first time. Ultimately, this should allow us to estimate their contribution to the ocean's global energy budget. However, several challenges must be addressed before the full potential of SWOT can be realized. Although the 21-day repeat cycle allows the 120-km swath to weave a global coverage, it results in poor temporal sampling. But small-scale processes evolve rapidly, so filling these sampling gaps is crucial for studying the energy budget of eddies and waves, as well as for applications in coastal sea-level change. This remains a substantial challenge for conventional data-assimilation approaches. Furthermore, at spatial scales less than 100 km, internal gravity waves, particularly internal solitary waves, can have larger SSHA amplitudes than the balanced motions of eddies at many times and places. Therefore, estimating surface-balanced velocities associated with (sub)mesoscale eddies directly from SWOT data becomes problematic when these two motions are comingled in the SSH snapshots. Separating a single SSH map into eddy and internal wave components is an underdetermined problem, requiring innovative physics-based algorithms[45,46]. Finally, submesoscale motions are clearly not in geostrophic balance; they are in gradient wind balance, which means that recovering their surface velocities from only SSHA requires solving the nonlinear momentum equations (see Methods).

One approach to address these challenges is to exploit the synergy between high-resolution SSH observations and other high-resolution satellite measurements of sea surface temperature, ocean interior data from the Argo float programme and dedicated field experiments using unmanned underwater vehicles such as gliders. Also, the SWOT and in situ measurements collected during the fast-sampling phase (March 2023–July 2023) provide higher-frequency information—SWOT is twice daily at crossover locations—offering a unique opportunity to test new hypotheses and methods.

Beyond these challenges lie the opportunities. For submesoscale turbulence, SWOT data could be used to estimate the kinetic energy flux between 100-km and sub-10-km scales, a fundamental process for understanding the ocean energy budget that is missing from our knowledge at present. One hypothesis to be tested is that the newly revealed energetic small-scale eddies are probably a product of the turbulent field of strongly interacting eddies covering a large range of scales, from 1 km to 500 km (refs. 42,47). We can also investigate the generation, propagation and dissipation of both linear and nonlinear internal waves. Such nonlocal energy propagation and dissipation are important elements of the tidal cycle but neither resolved nor effectively parameterized in present climate models. Also, although submesoscale eddies and internal waves have coherent signals in space and/or time, their interaction can create incoherent signals[48] and cascade energy fluxes between different spatial and temporal scales[49]. It is evident that SWOT observes both coherent structures and their interactions at sub-100-km scales (Figs. 1 and 3) owing to the extremely low instrument noise. SWOT presents a great opportunity to study these wave–eddy interactions globally.

Finally, SWOT provides measurements not discussed here, including terrestrial surface water[50] and other ocean features such as sea ice, sea surface roughness and the marine gravity field[51]. Extracting these signals globally from measurement noise and errors will require time, as many are not yet well understood in the context of SWOT SSH data and some were not expected to be so accurately captured. Altogether, this represents a wealth of information that will take years to explore and understand.

## Conclusions

Observing the oceans has always been challenging and expensive owing to their vastness and inaccessibility. However, satellite remote sensing has revolutionized modern oceanography. After three decades of using conventional altimeters to resolve mesoscale eddies, the SWOT mission heralds a new era in observing ocean variability at critical submesoscales less than 100 km, promising new discoveries and applications. In this study, we have showcased SWOT's capabilities with just two examples: submesoscale eddies and internal solitary waves, whose dynamics and energy have never been globally observed and are also poorly represented or entirely missed in low-resolution climate models. To take full advantage of the mission to generate new scientific insights, substantial efforts from the climate and data science communities are essential. These collaborative endeavours will refine advanced models, enhancing their predictive and diagnostic capabilities for the ocean, atmosphere and climate system.

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

## Methods

### SWOT data

This study uses data from 14 complete cycles of the SWOT science orbit, from 26 July 2023 to 8 May 2024. One complete cycle comprises 584 passes and takes 21 days to complete. We use two SWOT data products. (1) 'Level 2 KaRIn Low Rate Sea Surface Height Data Product – Unsmoothed', on a 250-m grid, version PIC0. The L2 product baseline C was produced by NASA and CNES (hereafter L2). (2) 'Level 3 sea surface height expert', on a 2-km grid, version 1.0, produced by the DUACS and SWOT Science Teams (hereafter L3)[52].

The L2 variable 'ssh_karin_2' was used in Figs. 1 and 3, which is the fully corrected SSH from KaRIn relative to the reference ellipsoid, using model-based wet troposphere correction (variables 'model_wet_tropo_cor' and 'sea_state_bias_cor_2'), as per recommendation from the SWOT project. We remove the mean sea surface (MSS; variable 'mean_sea_surface_cnescls' or 'CNES_CLS_2022'). We remove an along-track linear trend at each cross-track pixel; this removes the roll error but also some large mesoscale signal, but for our purpose of studying small-scale ocean dynamics in Figs. 1 and 3, this is not an issue. We do not apply quality control flags because we want to show all of the data. At certain times and locations, this unsmoothed 250-m gridded dataset contains long swell waves (for example, Fig. 1), which are filtered out in the 2-km product.

The L3 'ssha_unedited' was used in Figs. 2 and 4, which is the L2 smoothed SSHA 2-km product but with the L3 empirical calibration replacing the L2 calibration. This is important because L3 has been reconciled with the DUACS SSHA product to allow direct comparison and has been shown to be an improvement[53]. It should be noted that, although the empirical calibration might absorb a fraction of the difference between SWOT and DUACS, it will be small compared with other factors (for example, L4 smoothing)[52,54]. The SSH anomaly is from the MSS. We add back in the stationary internal tide SSH signal ('internal_tide_hret') that was removed as part of processing. We apply the variable 'quality_flag', which is used to identify and remove bad data, when computing the global variability (Fig. 2; all data points with a quality flag ≠ 0 are removed). We do not apply the flag for the study of the submesoscale eddy (Fig. 4) because it removes the core of the eddy as an outlier. Because SWOT coverage and accuracy is beyond the capability of the existing MSS product, there are residual small-scale bathymetric features in the SSHA data[55]. To correct for this, we compute a time mean for all cycles ($N = 14$), high-pass-filter it with a 50-km two-dimensional Gaussian kernel and subtract it from each pass. This process removes the small-scale bathymetric features from the SSHA while mostly bypassing the large-scale and mesoscale time mean. This is sufficient for our purposes, but it may remove a fraction of the ocean time mean signal; in the long run, a more accurate MSS product will be produced using the SWOT data[55].

### DUACS data

DUACS is the operational multimission production system for altimeter data developed by CNES/CLS (ref. 35). We use the DUACS near-real-time L4 dataset available from the Copernicus Marine Service. The data are provided daily on a 0.25° grid. We interpolate DUACS SSH onto SWOT grids at 2-km posting for the direct comparison in Fig. 2.

### Quantifying SWOT SSH beyond DUACS

To quantify the amount of variability seen by SWOT beyond DUACS (Fig. 2a), we compute the RMS in space (0.5° × 0.5°) and time (14 cycles; 26 July 2023 to 8 May 2024) of the difference between SWOT and DUACS SSHA. Figure 2b shows the RMS of 50-km high-passed SWOT SSHA only (filtering performed with a two-dimensional Gaussian kernel in both directions, although across swath, the window shortens towards the edges; we discard data 10 km from every edge).

This new view of small-scale ocean variability (Fig. 2) contains within it a wealth of information that will take years to precisely unpack, but which we know includes small-scale ocean currents and internal gravity waves (Figs. 3 and 4) and SWOT residual errors[32]−comprising residual instrument error including noise (for example, there are some noticeable track errors in the high latitudes) and residual geophysical signals not fully corrected for, including the wet troposphere effect, for which scales less than about 30 km are not well accounted for in model or radiometer measurements.

For the two case studies shown here (Figs. 3 and 4), the signal strength of each feature is $O(10\text{ cm})$. For context, the error standard deviation of SWOT at 6-km resolution is less than 0.5 cm. So, these examples are well above the noise floor. However, caution should be taken when analysing SWOT data globally, as both signal and errors are time- and space-dependent.

### (Cyclo)geostrophy and gradient wind

Mesoscale and submesoscale eddies are detected by satellite altimeters through SSHA, which serves as a proxy for surface pressure. SSHA is connected to eddy velocities ($U$) through the momentum equations, which incorporate nonlinear advection terms and Coriolis forces ($f$)[56]:

$$\mathbf{U}\nabla\mathbf{U} + f\mathbf{k} \times \mathbf{U} = -g\nabla\text{SSHA} \qquad (1)$$

in which $g$ is the acceleration owing to gravity and $\mathbf{k}$ is the vertical unit vector. When velocity gradients ($\nabla\mathbf{U}$) are much smaller than the Coriolis parameter $f$ (leading to $R \ll 1$, in which $R = \nabla\mathbf{U}/f$), eddies are in geostrophic balance, for which Coriolis forces counterbalance pressure forces. The geostrophic velocity is:

$$f\mathbf{k} \times \mathbf{U} = -g\nabla\text{SSHA} \qquad (2)$$

However, when velocity gradients approach or exceed $f$, the nonlinear terms become substantial, acting as centrifugal forces, similar to atmospheric hurricanes. This results in eddies characterized by a new dynamical state known as gradient wind balance[2,3,56,57]. Usually, large ocean eddies are in geostrophic balance and small eddies are in gradient wind balance. For an isolated eddy, such as the one depicted in Fig. 4 with a 15-km radius, in which the gradient of the geostrophic velocity is close to $f$, the flow cannot be considered in geostrophic balance. The gradient wind balance−called the cyclogeostrophic balance for an isolated circular eddy−can be denoted[56]:

$$V(r)^2/r + f V(r) = -g\text{SSHA}/r, \qquad (3)$$

in which $r$ is the radius of curvature. Solving for $V(r)$[56]:

$$V(r) = -fr/2 + (f^2 r^2/4 - g\text{SSHA})^{1/2}. \qquad (4)$$

With the Coriolis parameter $f = 10^{-4}\text{ s}^{-1}$, we found $V \approx 0.5\text{ m s}^{-1}$ for $r = 15$ km (instead of about $1\text{ m s}^{-1}$ for the geostrophic velocity). Note that the geostrophic Rossby number is Ro = 0.6 for this cyclone. Smaller eddies, such as those shown in Extended Data Fig. 2, exhibit larger magnitudes of the geostrophic velocity gradients, which can reach $2f$ or $3f$ and even higher. These characteristics suggest calling these eddies 'ocean hurricanes'.

A recent airborne experiment[47] uncovered a turbulent field of strongly interacting submesoscale eddies in gradient wind balance (equation (1)). In such eddy turbulence, the velocity gradients associated with these eddies involve not only the relative vorticity and the spin of the eddies but also the strain, which governs the stretching of an eddy by adjacent eddies, and divergence, which is linked to vertical motions. These gradients exhibit magnitudes larger than $f$ and play a critical role in the interactions of smaller eddies with larger eddies. The challenge for SWOT is recovering these gradients of equation (1) from only SSHA.

# Article

## Cautionary remarks on SWOT velocity

Our application of geostrophy and cyclogeostrophy with SWOT SSHA is predicated on two assumptions: (1) the SSHA signal observed by SWOT is primarily because of ocean currents and (2) the ocean currents can be explained by the simple steady-state balances. We are confident that, for the case study selected here (shown in Fig. 4), the first assumption holds true, based on the large amplitude of the eddy and concurrent observations from sea surface temperature and chlorophyll a imagery. When we compute velocity, we smooth the L3 SSHA field with a two-dimensional boxcar filter of 6 km (three grid points) before taking the horizontal gradients. This suppresses any residual noise resulting from high surface waves. Chelton et al.[58] used post-launch noise estimates to show that resolvable feature diameters are 8.5 km and larger in velocity. Our case study has a diameter of 30 km. We emphasize, however, that the first assumption is certainly not true for all SWOT-observed eddies, owing in large part to the entanglement of wave signals in the SSH at scales <100 km but also instrument measurement errors (for example, wet troposphere effect, sea state bias etc). The relative contribution of these factors is variable in space and time and is an area of active research. We show that the second assumption does not hold for geostrophy. We propose gradient wind/cyclogeostrophy as an improved approximation, as suggested by a recent airborne experiment[47] and literature[2].

## Vertical velocity

We estimate vertical velocity by tracking the time change of relative vorticity of an eddy over two successive SWOT passes (Extended Data Fig. 4). The dynamical balance between the change of relative vorticity and vertical stretching (or surface divergence with the minus sign) gives:

$$d\zeta/dt = (f + \zeta)dw/dz \qquad (5)$$

in which $\zeta$ is the relative vorticity, $f$ is the Coriolis parameter, $w$ is the vertical velocity and $z$ and $t$ are vertical and time coordinates, respectively. The time change of $\zeta$ is equal to the vertical stretching term. Assuming that the divergence is constant from the surface down to a given depth, $H$, we obtain an expression for the scale of vertical velocity, taking $w = 0$ at the surface, as:

$$w \approx -H(\Delta\zeta/\Delta t)/(f + \zeta) \qquad (6)$$

The area-averaged cyclogeostrophic relative vorticity associated with the Kuroshio small eddy—estimated to be half the geostrophic relative vorticity (see above)—is about $1f$ and changes by $+0.1f$ in 10.8 h. Using a range of mixed-layer depth values from the Argo climatology at the location of the eddy we observe here[59], from 50 to 125 m, we obtain a range of vertical velocity values from −6 to −14 m per day.

## Internal wave energetics

We follow refs. 39,40 to calculate the energy flux by internal solitary waves and tides. For linear waves, the dominant energy flux is the pressure work and can be calculated by $\int_{-H}^{0} u'p'dz$, in which $u'$ and $p'$ are the velocity and pressure perturbations associated with the internal waves, respectively. For nonlinear waves, the full energy flux includes two more terms, $\int_{-H}^{0} u'(K_e + P_e)dz$, in which $K_e$ and $P_e$ are the kinetic energy and available potential energy, respectively, of the nonlinear waves that can be calculated using wave SSH and stratification[60]. The stratification is used to estimate the vertical structure associated with the wave SSHA. The velocity fields used for kinetic energy can be derived from SSHA using dispersion relationships for inertial gravity waves and nonlinear solitary waves. The potential energy is associated with the vertical displacement of isopycnals that can be estimated by the vertical modal structures and SSHA. In the end, all energy flux can be calculated using SSHA and stratification. The details of the formulation can be found in previous studies[39,40,60].

Here we compute the energy flux of the internal solitary wave packet in the Andaman Sea observed by SWOT and shown in Fig. 3b,d. The mean vertical density profile and stratification is from the World Ocean Atlas 2023 climatology (WOA23). We tested using a total and seasonal mean profile and found no substantial difference. Using the observed SSHA peak amplitude of 20 cm (Fig. 3c) and the climatological mean density profile in a water depth of 700 m, we compute the internal solitary wave maximum vertical displacement, $\eta_0$, to be −39 m, its phase speed, $c_p$, as 2.26 m s$^{-1}$ and its half-width as 648 m. The depth-integrated kinetic energy and potential energy density are 42.9 and 31.5 kJ m$^{-2}$, respectively. The depth-integrated linear energy flux and the total energy flux at a peak amplitude of 20 cm are 140 and 176 kW m$^{-1}$, respectively. For waves with a peak amplitude of 10 cm, these numbers are reduced to 32 and 37 kW m$^{-1}$, respectively. To estimate a time average over one wave period, for this solitary wave packet of five waves generated every 12.4 h, the magnitude is reduced over each solitary wave hyperbolic secant profile by a factor of 3.5 and further reduced by a factor of 5 owing to the time gap without any nonlinear waves. This gives 8 kW m$^{-1}$ linear energy flux and 10 kW m$^{-1}$ total energy flux for a solitary wave packet of five waves of 20-cm peak amplitude and 1.8 kW m$^{-1}$ linear energy flux and 2 kW m$^{-1}$ total energy flux for the same wave packet but with 10-cm peak amplitude. The linear energy flux is more than twice as large as the previously estimated coherent $M_2$ tides[40] from altimetry, which were approximately 0.8 kW m$^{-1}$. The magnitudes of the total fluxes, ranging between 2 and 10 kW m$^{-1}$ from SWOT, are generally consistent with existing simulations in the Andaman Sea[61,62]. However, for the first time, this highly heterogeneous internal wave energy budget can be estimated at high spatial resolution from observations, which will contribute to the improvement of global ocean estimates.

## Data availability

All data used in this study are public. SWOT Level 2 data can be accessed through NASA (https://doi.org/10.5067/SWOT-SSH-2.0) or AVISO (https://doi.org/10.24400/527896/a01-2023.016). The SWOT Level 3 data are available through AVISO (https://doi.org/10.24400/527896/a01-2023.018). The DUACS gridded data are accessible through CMEMS (https://doi.org/10.48670/moi-00149). Hydrographic profiles from World Ocean Atlas (WOA23) are available at https://www.ncei.noaa.gov/products/world-ocean-atlas. Mixed-layer depth data are available at https://mixedlayer.ucsd.edu/. Simulated SSH data from the MITgcm LLC4320 data are available at https://doi.org/10.5067/KARIN-2MES1. Ocean colour and sea surface temperature imagery was produced by NASA's Ocean Biology Processing Group, available at https://oceandata.sci.gsfc.nasa.gov/. Analysis and figures were coded in Matlab. We use M_Map, a mapping package for Matlab, available at www.eoas.ubc.ca/~rich/map.html. We plot bathymetry using the ETOPO11 Arc-Minute Global Relief Model from the NOAA National Geophysical Data Center available at https://www.ncei.noaa.gov/access/metadata/landing-page/bin/iso?id=gov.noaa.ngdc.mgg.dem:316. We use 'cmocean' colour bars developed by Kristen Thyng and colleagues (https://matplotlib.org/cmocean/) and 'brewermap' colour bars developed by Cynthia Brewer and colleagues (https://colorbrewer2.org/).

## Code availability

The code used to generate the figures in this article is available at https://doi.org/10.5281/zenodo.14736001 (ref. 63).

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

**Acknowledgements** The research was carried out at the Jet Propulsion Laboratory, California Institute of Technology, under a contract with the National Aeronautics and Space Administration (80NM0018D0004). M.A., J.W., P.K. and L.-L.F. are supported by the SWOT mission. SWOT is a flagship international mission and the culmination of two decades of work by over a thousand individuals across several academic generations. This paper is a tribute to the remarkable team from NASA, CNES, CSA and the UK Space Agency, whose collective expertise and commitment brought this mission to fruition. We thank Z. Zhao for providing the precise value of the linear $M_2$ tidal flux in the Andaman Sea. We thank the editor and three reviewers whose input improved the manuscript.

**Author contributions** Conceptualization: L.-L.F., P.K. Methodology: P.K., J.W., M.A. Formal analysis and investigation: M.A. Data curation: M.A., G.D. Writing – original draft: M.A., J.W. Writing – review and editing: M.A., J.W., P.K., L.-L.F., G.D. Visualization: M.A. Project administration and financing acquisition: L.-L.F.

**Competing interests** The authors declare no competing interests.

**Additional information**
**Correspondence and requests for materials** should be addressed to Matthew Archer or Jinbo Wang.

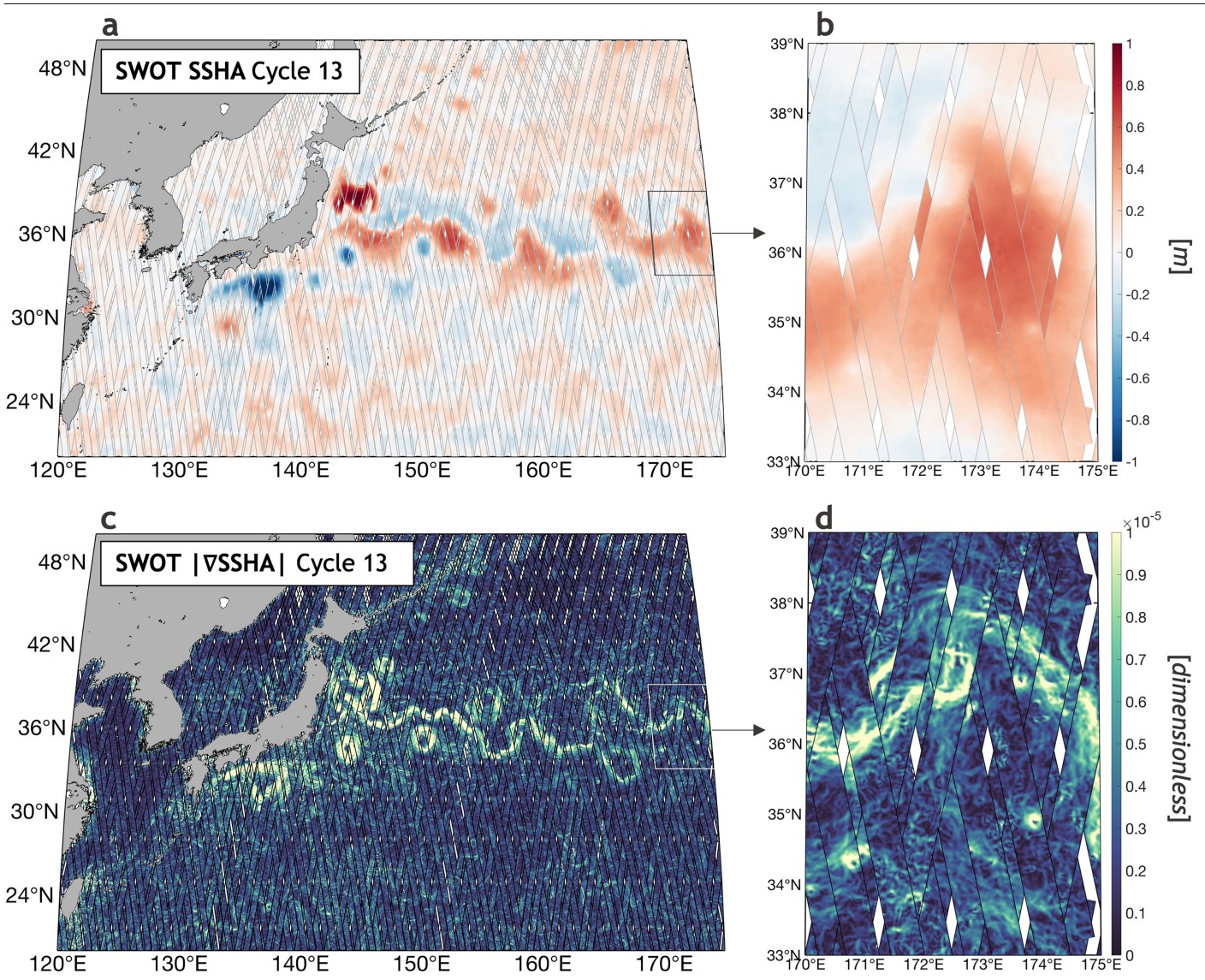

**Extended Data Fig. 1 | SWOT observes the Kuroshio Current jet and submesoscale instabilities. a**, SWOT SSHA from cycle 13 over the Kuroshio Extension region in the northwest Pacific. **b**, Zoom-in. **c**, SSHA spatial gradient magnitude. **d**, Zoom-in to the same area as **b**. Eddies with diameters of approximately 20 km and small-scale filament structures are distinctly identifiable in **d**.

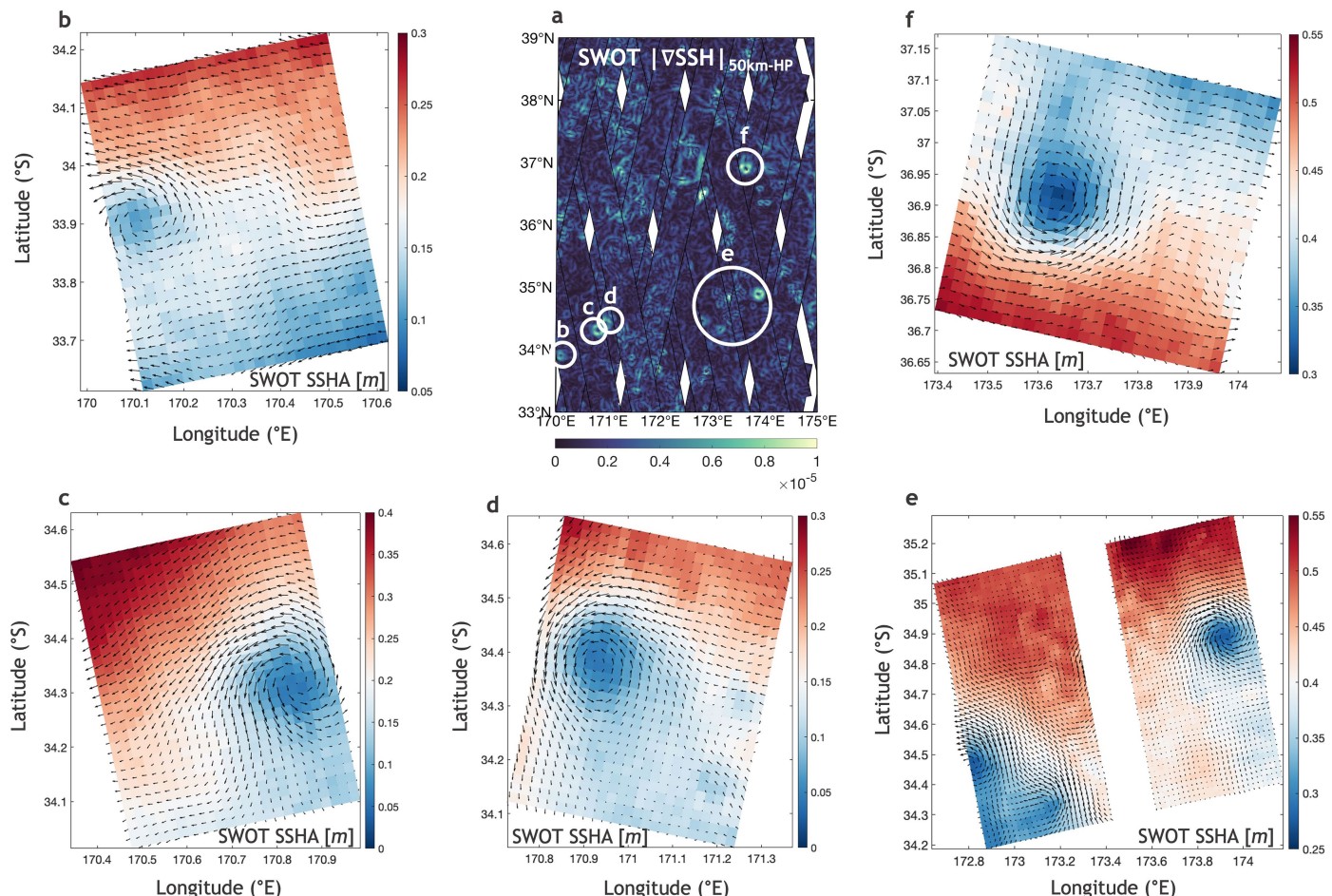

**Extended Data Fig. 2 | Submesoscale coherent eddies captured by SWOT in the Kuroshio Extension. a**, SSHA spatial gradient magnitude, high-pass-filtered with a 50-km two-dimensional Gaussian kernel, superimposed with white circles that identify the five example eddies plotted in panels **b**–**f**. Panels **b**–**f** show SWOT total SSHA superimposed with black geostrophic velocity vectors.

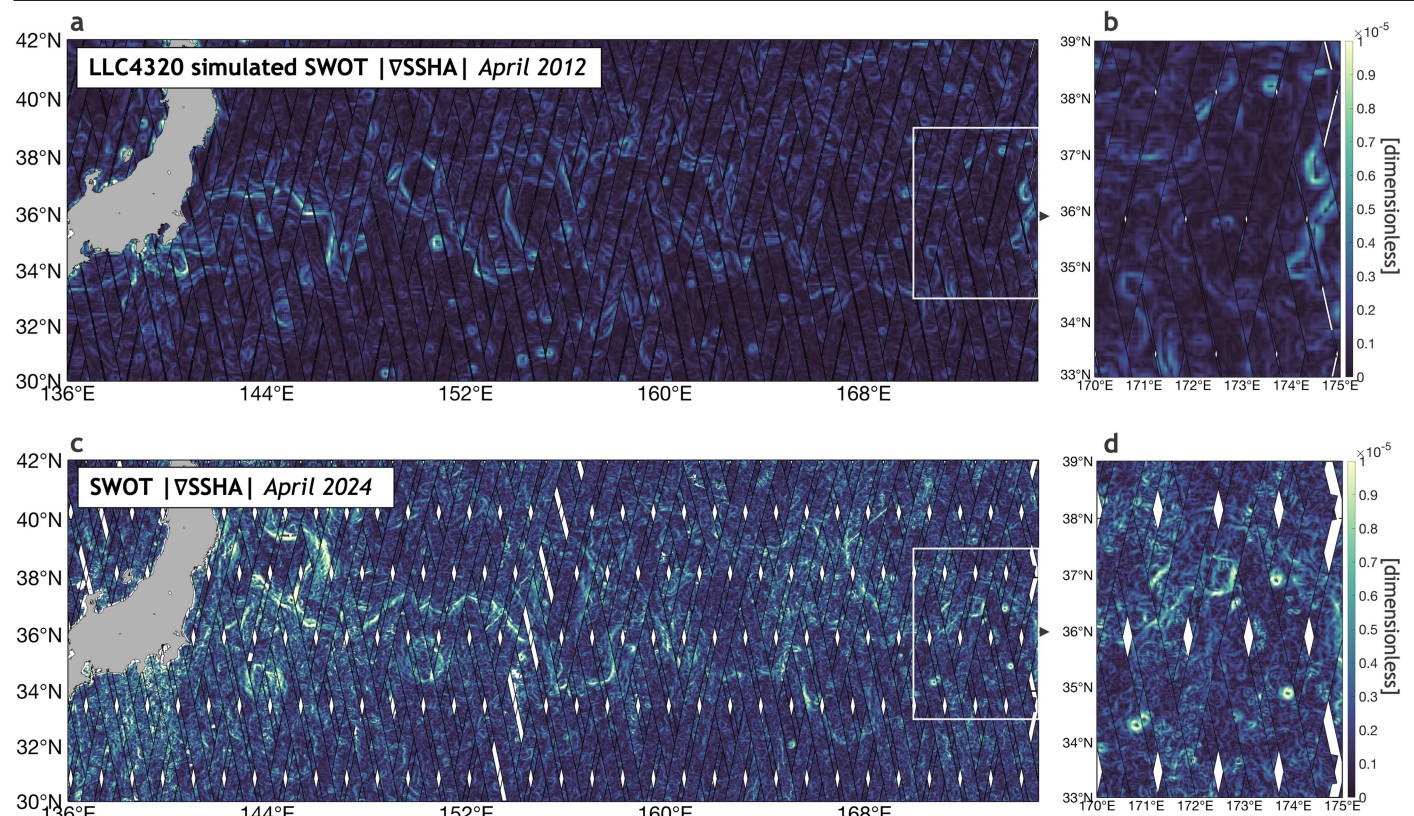

**Extended Data Fig. 3 | SWOT observes much stronger SSHA gradients than models predict in the Kuroshio Extension. a**, One cycle of simulated SWOT SSHA gradient magnitude from the 1/48° global LLC4320 model, high-pass-filtered with a 50-km two-dimensional Gaussian kernel, during April 2012. **b**, Zoom-in to a smaller region. **c**, Cycle 13 of SWOT SSHA gradient magnitude, high-pass-filtered with a 50-km two-dimensional Gaussian kernel. **d**, Zoom-in to the same region as **b**.

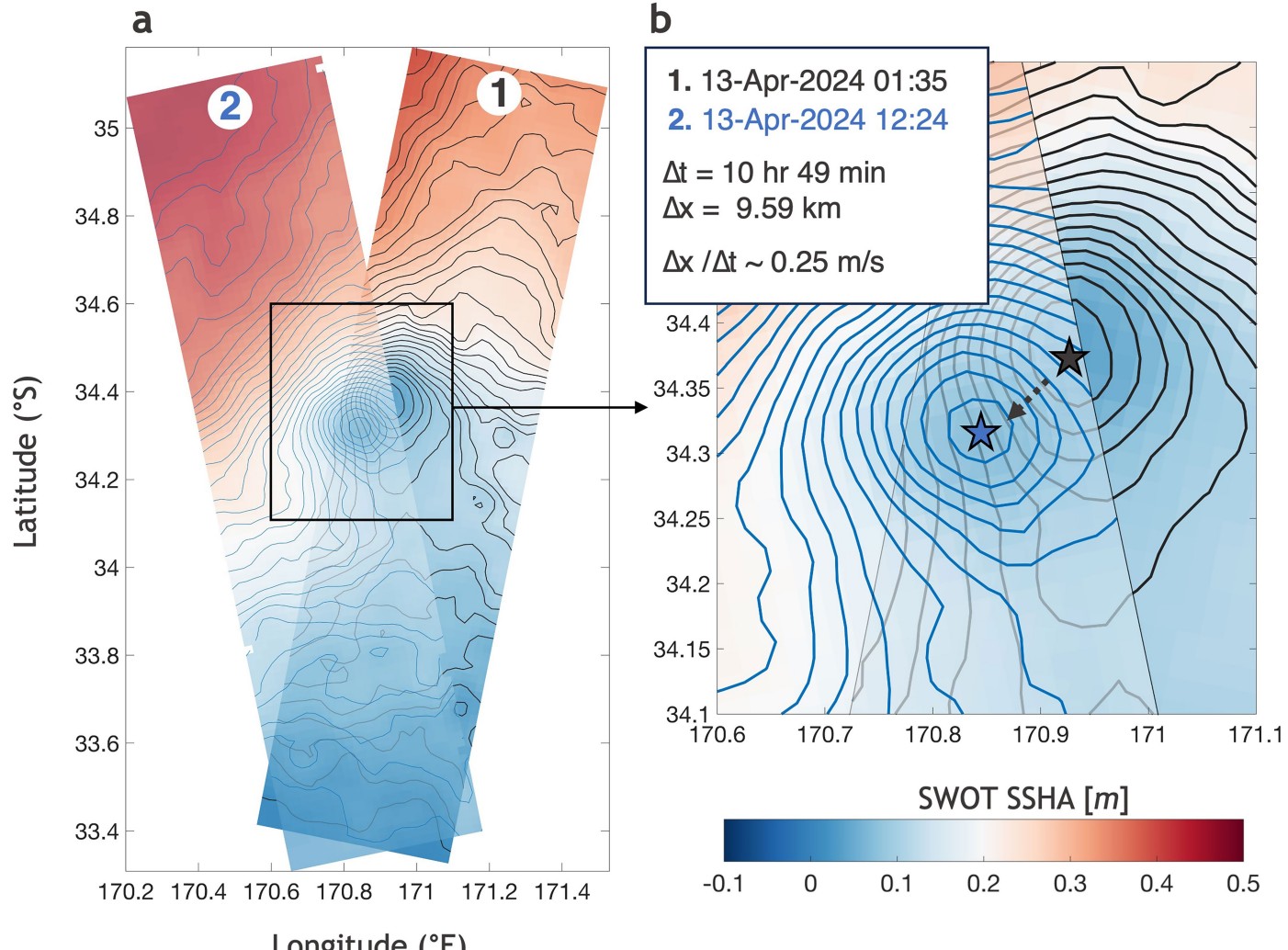

**a**

**b**

**1.** 13-Apr-2024 01:35
**2.** 13-Apr-2024 12:24

Δt = 10 hr 49 min
Δx = 9.59 km

Δx /Δt ~ 0.25 m/s

SWOT SSHA [*m*]

**Extended Data Fig. 4 | A submesoscale eddy is observed twice by SWOT within 12 h. a**, An ascending and descending pass of SWOT SSHA during cycle 13 of the science orbit, showing an eddy captured twice in 10.8 h. **b**, Zoom-in to the eddy centres showing the distance of 9.59 km that the eddy centre travelled in 10.8 h, which represents a linear translation speed of 0.25 m s⁻¹.