## [Peer Review File · Nature]

Wide-swath satellite altimetry unveils global submesoscale ocean dynamics

Corresponding Author: Dr Matthew Archer

Version 1:

Reviewer comments:

Referee #1

(Remarks to the Author)

Recommendation: accept after revision

Summary

The manuscript highlights the quality of oceanographic data currently streaming from NASA's SWOT satellite -- particularly its wide-swath, high resolution measurements of SSH from its KaRIn interferometer. The authors emphasize and convincingly demonstrate that the resolution of SSH is far higher than expected from its engineering requirements. Focusing on three startling images, they discuss SWOT's promise for global analysis of internal waves and submesoscale dynamics.

Analysis

This is a timely and convincing demonstration of the SSH data being provided currently by SWOT. It is of great use to the oceanographic community, and will surely be appreciated by the broad readership of Nature.

The figures are beautiful, expertly done, and should not be changed. The text of the paper, however, needs some improvement, and few crucial points and challenges are glossed over or hidden from view.

Major point:

The authors do not sufficiently address an evident disconnect from the presentation. On the one hand, they rightly point out that SWOT is revealing SSH features down to scales of $O(1\text{km})$ (or less in the case of swell in Fig 1!!), and that the features they are seeing have Rossby numbers $\sim O(1)$, and therefore are not geostrophic. Yet, throughout the paper, the availability of this high-quality submesoscale SSH data is tacitly assumed to provide the underlying velocities needed to compute things like submesoscale kinetic energy, and vertical fluxes of heat.

In the case of the internal, swell, and solitary wave data they highlight in Figs. 1 and 3 (waves that seem almost monochromatic, un-Doppler-shifted, and lightly refracted), it is very reasonable to assume these features directly reflect the surface signature of a known dispersion relationship, and thus one can confidently say something about their velocities and energy. In the case of the submesoscale vortex considered in Fig 4, the discussion in lines 153-170 at least tries to get at this, suggesting in a confusing way that cyclogeostrophic balance may be a better model. Perhaps, but this is not well demonstrated in the recent literature on submesoscales. Moreover, the feature in Fig 4 is fairly large relative to what SWOT can see.

I am more concerned with these statements in lines...

184-186: that SWOT can be used to derive vertical velocities (implied to be the vertical velocities needed to constrain submesoscale vertical heat flux): the vertical velocities needed are likely those associated with highly-nongeostrophic fronts, features that are also not captured by cyclogeostrophy. One must also confidently filter the wave signal first

195-197: that SWOT can be used to estimate submesoscale lateral kinetic energy fluxes (similar issues as described in

previous point).

My suggestion is to simply be more upfront that, while there is incredible promise, and no one can argue with the data quality, the science is not quite there to accurately compute some of the quantities the authors claim are on the near horizon.

Specific points:

58-64: It is implied without caveat that SWOT will allow us to measure these vertical fluxes...

153-170: The authors should include the term "cyclotrophic balance" somewhere here so that interested readers know what to search for. Also, they should more honestly discuss what's known, and that this may not work for smaller scale features, or in other regions, etc. It comes across as if this is a solved problem, just as accurate as using geostrophy with AVISO.

182-183: need citation for claim that submesos heat flux is 5-10x that due to mesoscale.

184-185: Again, caveats are needed. This will not get the vertical velocities needed for the submesoscale vertical fluxes of heat.

195-97: Add caveats

223-224: Even with waves filtered, QG is geostrophic.

Referee #2

(Remarks to the Author)

This work presents an outstanding demonstration of the technical capabilities of the SWOT satellite to tackle open questions in physical oceanography on the role of small-scale processes in the climate system. For the very first time, SWOT enables us to observe small structures globally and quantify the amplitude of solitary internal tides and ubiquitous small-scale eddies. This greatly advances our knowledge of small-scale ocean dynamics and helps parametrize climate models in view of more accurate climate projections.

With clear references to the building blocks of this work, the authors provide robust and reproducible conclusions.

The approach and methodology are valid, and data from SWOT show unprecedented results. Comments on the data products used:

- Figures 2 and 3 are produced with the L3 ssha_unedited from the V1.0 data product. This SSHA is not filtered or edited. Averaging over 14 cycles may be enough to remove the KaRIn noise field, but there could still be residuals. Is this accounted for? Does the residual noise impact the results of Figures 2 and 3? Can the authors please comment on this?
- The L2 product is referred to as "karin_ssh_2" in the SWOT data. Is it the "ssh_karin_2"? If so, please correct.

Suggested improvements:

- Generally, SWOT's impressive technical capability and interest are clear from the examples shown. However, some physical interpretation is missing to demonstrate SWOT's potential to the community fully. For instance, in the two quantitative examples shown:
 - 1) Why is it important to gain this knowledge on the vertically integrated time-averaged linear energy flux of solitary waves? What do these values tell us about the dynamics in the region?
 - 2) What does this kind of small eddy represent in this region? What is the benefit of observing them with SWOT? Are they always observable? What would be the long-term physical consequences of following the evolution of such structures with SWOT?
- Line 101: You mention that swells were not expected to be observed, but part of the surface long swell wave signals pass through the filter due to the high signal-to-noise ratio of KaRIn. Is this a specificity of this region? Are there regions where the signal-to-noise is not high enough? What can we learn from the knowledge of the swell? Why is it important?
- Line 109: Can you extend the explanation on why the pattern of the RMS differences in Fig 2 indicates that the variability is related more to ocean physics than measurement noise or errors? Is this the case also in less energetic regions?
- Fig2: The titles of the two panels mention SSH. Is it SSH or SSHA?
- Fig2b shows the SSH variability for scales smaller than 50 km. What is SWOT observability in this case? Down to which

scales do the authors interpret the results? Chelton (2024) shows a Postlaunch Update on the Effects of Instrumental Measurement Errors on SWOT Estimates of Sea Surface Height, Velocity, and Vorticity (10.1175/JTECH-D-24-0035.1). How does this compare to the results in the regions investigated here?

- Line 146: is the variability of the energy and energy fluxes investigated in this work? Do you refer to spatial or temporal variability?

- Line 149: have these magnitudes been observed with other models or data? Validating these fluxes is fundamental to correctly interpreting the small-scale SWOT signals. I understand that the validation of the structures shown is not the scope of this work, but the authors should comment on the importance of such a process.

- Fig 4: the image refers to cyclogeostrophy, which is explained later in the methods but not mentioned in the main text. Around lines 161 to 168, I suggest you mention that you are describing the cyclogeostrophic balance, referencing the methods section.

- Internal wave energetics (lines 317 and 319): please check if you want to refer to Figure 3 instead of Figure 4.

- Opportunity and challenges of SWOT:

The KaRIn noise is indeed low, but even after de-noising, some residual KaRIn noise may still be present on the swaths. This may not affect the SSH much, but accessing higher-order eddy diagnostics (velocity, vorticity) could still be challenging, and the interpretation of such quantities at small scales should be done with caution. I suggest including a comment on this as part of the challenges.

Referee #3

(Remarks to the Author)

This paper reviews the capabilities of the recently launched (Surface Water and Ocean Topography) SWOT satellite in observing fine-scale ocean dynamics, which will significantly advance our understanding of submesoscale ocean processes in ways that conventional altimetry could not. The ocean processes of scales 1-100km play crucial roles in vertical transport and energy transfer, impacting broader ocean circulation, biodiversity, and climate. Here, the authors present some of the first wide-swath satellite altimetry observations and exploit them to provide a first global statistical view of the variability at those scales. They also demonstrate how SWOT data can be used to analyse the dynamics and energy of submesoscale eddies or internal solitary waves, unveiling significant energy fluxes and dynamics often underestimated by simulations.

The first part of the paper is very valuable, as it presents how wide-swath altimetry is revolutionizing ocean topography observations, particularly by unveiling fine-scale variability that conventional altimetry systems like DUACS cannot. Unsmoothed SWOT swaths can detect much finer structures, such as filaments, internal waves, and ocean surface swell waves. By comparing SWOT data to DUACS, the authors reveal significant differences in the observed variability, especially in regions with high kinetic energy or where internal gravity waves are generated. The second part details two case studies that illustrate how SWOT data can be used to calculate the dynamics and energy of nonlinear solitary waves and submesoscale eddies. In the case of internal solitary waves, the total energy flux is stronger than coherent M2 internal tides in this region. In the case of submesoscale eddies, SWOT confirms that numerical simulations underestimate their vorticity and associated vertical velocities, minimizing their impact on the ocean dynamics and heat budget. While the methodology for the internal waves is largely based on Qiu et al. (2024), this paper offers a more accessible and comprehensive explanation of SWOT's application, easier for a broader audience to understand and use.

Given the importance of ocean submesoscale processes across multiple fields, I am confident that these examples will be widely used by researchers in various domains. It will simplify the understanding of the significant advances provided by SWOT, enabling those new to SWOT data or unfamiliar with its development to grasp its potential and begin exploring basic applications. With that in mind, the paper could benefit from some instructions, including on the necessary precautions needed to handle this data. I include some suggestions into the general comments.

The paper is very well written. The abstract is appropriate, and the introduction is valuable in highlighting the role of the submesoscale dynamics in the climate system and showing the limits of past observations of these small scale features. The presentation is sound and the figures are clear. The conclusions are valid. SWOT heralds a new era in observing ocean variability at submesoscale, and this paper clearly supports this point.

General Comments

The challenge of applying (quasi)geostrophic theories directly to SWOT SSH snapshot is mentioned at the end of the paper, but not developed. How does that impact the submesoscale eddy example? Is it fixed by applying the cyclogeostrophic balance? Which precautions should one take before applying the same method on another submesoscale eddy? The paper would benefit from further instructions on when similar computation could be applied or not.

Similarly, how is the SSH noise impacting the test cases? Chelton et al. (2022) anticipated that additional smoothing would be required for most applications of SWOT estimates of velocity and vorticity. Has any of this additional smoothing been applied here, and when should it be considered when reproducing this analysis on other submesoscale eddies?

There is no mention of measurement, geophysical correction, or mapping uncertainty in the paper. Similarly, the Sea

Surface Height 1d plots in Figures 3c and 4e show no error bars or confidence intervals. Having a broad idea of the order of magnitude of this uncertainty would be valuable to the reader (How big is it? Is it the same within the whole swath?)

Other comments:

L 89: "new small-scale features". Not really new but detectable for the first time.

L94: The figure does not clearly show the "smaller-scale eddies" mentioned here.

L 96: The wavelength is 30-km in the figure.

L 101-106: The conclusion from is a bit unclear to me. While it shows the impressive capability of SWOT in detecting small scale signals of only hundreds of meters, it also mentions that this signal was supposed to be filtered out. Does that imply that the user should filter himself this signal when looking at larger scale (1-100km) processes? Is it totally corrected in the L3 dataset?

L112: RMS or STD?

L 115-116: "20% of the total SSHA RMS" Where does this percentage come from? This needs to be complemented with a map of the total SSHA RMS or a paper showing one should be added here.

L129: "30% of SSHA variability"

L145-146: "but the fundamental question for ocean energetics can only be answered by measuring their amplitude".

L 158: "warmer". It looks colder from the Figure.

L 168-170: Reference?

L174-176 and Extended Data Fig. 3. I am not sure the high-resolution model has been subsampled on SWOT observations. Does the subsampling impact the RMS SSH values? It seems to me that the comparison would make more sense if the SWOT values were compared with the direct high-resolution model outputs.

L185-186: Refer to Figure

Extended Data 2: Arrows?

References 27 and 41 might have titles swapped. I was not able to access them.

L251: What is the grid resolution of the L3 dataset?

L267: It would be nice to have the time period over which this standard deviation has been computed.

L317: I think you're referring to Fig. 3.

L319: same as the previous comment

Version 2:

Reviewer comments:

Referee #1

(Remarks to the Author)

I am satisfied with the authors' revisions and recommend publication.

Referee #2

(Remarks to the Author)

Thank you for your work. The new version of the manuscript answers the comments raised in the review and should be published.

Referee #3

(Remarks to the Author)

The authors addressed all of my suggestions and questions and I have no further comments.

Response to Reviewers

Blue: author's responses to reviewers.

Referee #1 (Remarks to the Author):

Recommendation: accept after revision

Summary

The manuscript highlights the quality of oceanographic data currently streaming from NASA's SWOT satellite -- particularly its wide-swath, high resolution measurements of SSH from its KaRIn interferometer. The authors emphasize and convincingly demonstrate that the resolution of SSH is far higher than expected from its engineering requirements. Focusing on three startling images, they discuss SWOT's promise for global analysis of internal waves and submesoscale dynamics.

Analysis

This is a timely and convincing demonstration of the SSH data being provided currently by SWOT. It is of great use to the oceanographic community, and will surely be appreciated by the broad readership of Nature.

The figures are beautiful, expertly done, and should not be changed. The text of the paper, however, needs some improvement, and few crucial points and challenges are glossed over or hidden from view.

Thank you for your comments that have brought accountability and precision to the paper.

Major point:

The authors do not sufficiently address an evident disconnect from the presentation. On the one hand, they rightly point out that SWOT is revealing SSH features down to scales of $O(1\text{km})$ (or less in the case of swell in Fig 1!!), and that the features they are seeing have Rossby numbers $\sim O(1)$, and therefore are not geostrophic. Yet, throughout the paper, the availability of this high-quality submesoscale SSH data is tacitly assumed to provide the underlying velocities needed to compute things like submesoscale kinetic energy, and vertical fluxes of heat. In the case of the internal,

swell, and solitary wave data they highlight in Figs. 1 and 3 (waves that seem almost monochromatic, un-Doppler-shifted, and lightly refracted), it is very reasonable to assume these features directly reflect the surface signature of a known dispersion relationship, and thus one can confidently say something about their velocities and energy. In the case of the submesoscale vortex considered in Fig 4, the discussion in lines 153-170 at least tries to get at this, suggesting in a confusing way that cyclogeostrophic balance may be a better model. Perhaps, but this is not well demonstrated in the recent literature on submesoscales. Moreover, the feature in Fig 4 is fairly large relative to what SWOT can see.

We have edited the writing to emphasize the entanglement of waves and currents in the SSH field, which as the reviewer points out, can contaminate the analysis of each. For the examples we present, we are confident the process we focus on in each example (wave or current) explains the majority of SSH variance in the swath, and write

“...we present two examples: an internal solitary wave packet and a submesoscale coherent eddy. SWOT captures the SSH signatures of such features in abundance, yet they are often comingled in space; the following examples were chosen for the singular spatial dominance of waves or currents in the SSH field (see Methods).” [Lines 147-150]

We also make clear that simply computing geostrophic velocity of SSH with waves present is absurd:

“If one were to compute the geostrophic currents directly from these observations without removing such features, it would yield a current speed of 12 m/s!” [Lines 154-156]

We have created a new section in the Methods titled “*Cautionary remarks on SWOT velocity*” to highlight the assumptions made when computing these quantities (e.g., velocity and vorticity) and emphasized the caveats relating to the use of SSH to derive velocity.

*“**Cautionary remarks on SWOT velocity.** Our application of geostrophy and cyclogeostrophy with SWOT SSHA is predicated on two assumptions: (i) the SSHA signal observed by SWOT is primarily due to ocean currents, and (ii) the ocean currents can be explained by the simple steady state balances. We are confident that for the case study selected here (shown in Figure 4) the first assumption holds true, based on the large amplitude of the eddy, and concurrent observations from SST and chlorophyll-a imagery. When we compute velocity, we smooth the L3 SSHA field with a 2D boxcar filter of 6 km (3 grid points) before taking the horizontal gradients. This suppresses any*

residual noise due to high surface waves. Chelton (2024)⁵⁷ used postlaunch noise estimates to show resolvable feature diameters are 8.5 km and larger in velocity. Our case study has a diameter of 30 km. We emphasize, however, that the first assumption is certainly not true for all SWOT-observed eddies, due in large part to the entanglement of wave signals in the SSH at scales <100 km, but also instrument measurement errors (e.g., wet troposphere effect, sea state bias, etc.) The relative contribution of these factors is variable in space and time and is an area of active research. We show the second assumption does not hold for geostrophy. We propose gradient wind/cyclogeostrophy as an improved approximation, as suggested by a recent airborne experiment⁴⁷ and literature².” [Lines 530-545]

... and in the Main Text:

“...which is a clear indication the eddy is not in geostrophic balance (for which $R \ll 1$, see Methods). Given the small radius and large SSHA amplitude of this eddy, this is not surprising. A better approximation to compute the velocity of this eddy is cyclogeostrophy, which adds a third term to the force balance: centrifugal acceleration. This term is needed because the strong curvature of the currents around the low-pressure center requires an acceleration to maintain balance....” [Lines 177-182]

We have improved the description of gradient wind/cyclogeostrophic balance in the Methods...

“(Cyclo)geostrophy and gradient wind. Mesoscale and submesoscale eddies are detected by satellite altimeters through SSHA, which serves as a proxy for surface pressure. SSHA is connected to eddy velocities (U) through the momentum equations, which incorporate nonlinear advection terms and Coriolis forces (f)⁵⁵:

$$\mathbf{U} \cdot \nabla \mathbf{U} + f \mathbf{k} \times \mathbf{U} = -g \nabla SSHA \quad (1)$$

where g is the acceleration due to gravity, and \mathbf{k} is the vertical unit vector. When velocity gradients (∇U) are much smaller than the Coriolis parameter f (leading to $R \ll 1$, where $R = \nabla U / f$), eddies are in geostrophic balance, where Coriolis forces counterbalance pressure forces. The geostrophic velocity is diagnosed as:

$$f \mathbf{k} \times \mathbf{U} = -g \nabla SSHA \quad (2)$$

However, when velocity gradients approach or exceed f , the nonlinear terms become significant, acting as centrifugal forces, similar to atmospheric hurricanes. This results in eddies characterized by a new dynamical state known as gradient wind balance^{2,22,55,56}. Usually, large ocean eddies are in geostrophic balance and small eddies in gradient wind balance. For an isolated eddy, such as the one depicted in Figure 4 with a 15 km

radius, where the gradient of the geostrophic velocity is close to f , the flow cannot be considered in geostrophic balance. The gradient wind balance – called the cyclogeostrophic balance for an isolated circular eddy – can be diagnosed as⁵⁵:

$$\frac{V(r)^2}{r} + fV(r) = -g\frac{SSHA}{r}, \quad (3)$$

where r is the radius of curvature. Solving for $V(r)$ ⁵⁵:

$$V(r) = \frac{-fr}{2} + \left(\frac{f^2 r^2}{4} - g.SSHA \right)^{1/2}. \quad (4)$$

With the Coriolis parameter equal to $f = 10^{-4} s^{-1}$, we found $V \sim 0.5$ m/s for $r = 15$ km (instead of ~ 1 m/s for the geostrophic velocity). Note that the geostrophic Rossby number is $R=0.6$ for this cyclone. Smaller eddies, like those displayed in Extended Data Fig. 2, exhibit larger magnitudes of the geostrophic velocity gradients, which can reach 2 or 3 f and even higher. These characteristics suggest calling these eddies “ocean hurricanes”.

A recent airborne experiment⁴⁷ uncovered a turbulent field of strongly interacting submesoscale eddies in gradient wind balance (Equation 1). In such eddy turbulence, the velocity gradients associated with these eddies involve not only the relative vorticity, the spin of the eddies, but also the strain, which governs the stretching of an eddy by adjacent eddies, and divergence, which is linked to vertical motions. These gradients exhibit magnitudes larger than f and play a critical role in the interactions of smaller eddies with larger eddies. The challenge for SWOT is recovering these gradients of Equation 1 from only SSHA.” [Lines 487-528]

I am more concerned with these statements in lines...

184-186: that SWOT can be used to derive vertical velocities (implied to be the vertical velocities needed to constrain submesoscale vertical heat flux): the vertical velocities needed are likely those associated with highly-nongeostrophic fronts, features that are also not captured by cyclogeostrophy. One must also confidently filter the wave signal first.

In the Methods section “(Cyclo)geostrophy and gradient wind” we now distinguish what we call gradient wind balance, that explicitly includes the divergence, and the so-called cyclogeostrophic balance (usually associated with isolated eddies) that does not include divergence (see the new Methods section). The cyclogeostrophic balance is only the balance of isolated circular eddies as detailed above. We now mention at the end of this section:

“A recent airborne experiment⁴⁷ uncovered a turbulent field of strongly interacting submesoscale eddies in gradient wind balance (Equation 1). In such eddy turbulence,

the velocity gradients associated with these eddies involve not only the relative vorticity, the spin of the eddies, but also the strain, which governs the stretching of an eddy by adjacent eddies, and divergence, which is linked to vertical motions.” [Lines 522-526]

This means that, solving the nonlinear Equation 1 leads to recovery of the divergence. Note that at the end of this section, we mention: “The challenge for SWOT is recovering these gradients of Equation 1 from only SSHA.” [Lines 527-528]

Another way to recover the divergence is to track the Lagrangian advection of a particular eddy, as detailed in the Methods section “Vertical velocity”. Thus, the Methods section now explicitly mentions that SWOT observations can only get access to the divergence at the surface. Then, the vertical velocity at a given depth is diagnosed assuming the divergence is constant down to this depth. We have written in the Main Text:

“One can theoretically use SWOT observations to derive vertical velocities at a given depth from the surface divergence, assuming this divergence is constant down to this depth (see Methods): we estimate the vertical velocity for an eddy in the Kuroshio Extension that was observed by SWOT twice in 12-hours...” [Lines 202-205]

This assumption related to the divergence is also explicitly mentioned in the Methods section “Vertical velocity”:

“Assuming the divergence is constant from the surface down to a given depth, H , we obtain an expression for the scale of vertical velocity, taking $w = 0$ at the surface, as...” [Lines 556-557]

195-197: that SWOT can be used to estimate submesoscale lateral kinetic energy fluxes (similar issues as described in previous point).

See below*

My suggestion is to simply be more upfront that, while there is incredible promise, and no one can argue with the data quality, the science is not quite there to accurately compute some of the quantities the authors claim are on the near horizon.

***We have rearranged and re-worded some of the section ‘Opportunities and Challenges...’ to give more emphasis to the challenges:**

“SWOT’s precise SSH measurements of ocean submesoscale processes enable us to quantify their dynamics for the first time. Ultimately, this should allow us to estimate their contribution to the ocean’s global energy budget. However, several challenges must be

addressed before the full potential of SWOT can be realized. While the 21-day repeat cycle allows the 120-km swath to weave a global coverage, it results in poor temporal sampling. But small-scale processes evolve rapidly, so filling these sampling gaps is crucial for studying the energy budget of eddies and waves, as well as for applications in coastal sea level change. This remains a significant challenge for conventional data assimilation approaches. Furthermore, at spatial scales below 100 km, internal gravity waves, particularly internal solitary waves, can have larger SSHA amplitudes than the balanced motions of eddies at many times and places. Therefore, estimating surface balanced velocities associated with (sub)mesoscale eddies directly from SWOT data becomes problematic when these two motions are comingled in the SSH snapshots. Separating a single SSH map into eddy and internal wave components is an underdetermined problem, requiring innovative physics-based algorithms^{45,46}. Finally, submesoscale motions are clearly not in geostrophic balance; they are in gradient wind balance, which means that recovering their surface velocities from only SSHA requires solving the nonlinear momentum equations (see Methods).” [Lines 212-228]

We have also included more caveats in the Methods, such as the new section of cautionary remarks (see above).

Specific points:

58-64: It is implied without caveat that SWOT will allow us to measure these vertical fluxes...

Here we introduce the current understanding of ocean submesoscales; we are not discussing SWOT, which is not introduced until the next paragraph.

153-170: The authors should include the term "cyclotrophic balance" somewhere here so that interested readers know what to search for. Also, they should more honestly discuss what's known, and that this may not work for smaller scale features, or in other regions, etc. It comes across as if this is a solved problem, just as accurate as using geostrophy with AVISO.

We have included the definition of cyclogeostrophy...

“A better approximation to compute the velocity of this eddy is cyclogeostrophy, which adds a third term to the force balance: centrifugal acceleration. This term is needed because the strong curvature of the currents around the low-pressure center requires an acceleration to maintain balance.” [Lines 179-182]

...and re-worded to avoid the implication this is a solved problem...

“Velocity can be derived from SSH using the geostrophic approximation (with strong precautions, see Methods) – the predominant force balance of the ocean between the pressure gradient force and the effect of Earth’s rotation – and it exceeds 1-m/s. However, the Rossby number...” [Lines 172-175]

182-183: need citation for claim that submeso heat flux is 5-10x that due to mesoscale. **Added Su et al. (2018).**

184-185: Again, caveats are needed. This will not get the vertical velocities needed for the submesoscale vertical fluxes of heat.

See reply above.

195-97: Add caveats

We’ve rearranged and re-worded some of the section ‘Opportunities and Challenges...’ so that we say *after* addressing issues, we would be able to compute e.g. kinetic energy flux (see Main Text).

223-224: Even with waves filtered, QG is geostrophic.

We have modified this statement and now write:

“Furthermore, at spatial scales below 100 km, internal gravity waves, particularly internal solitary waves, can have larger SSHA amplitudes than the balanced motions of eddies at many times and places. Therefore, estimating surface balanced velocities associated with (sub)mesoscale eddies directly from SWOT data becomes problematic when these two motions are comingled in the SSH snapshots. Separating a single SSH map into eddy and internal wave components is an underdetermined problem, requiring innovative physics-based algorithms^{45,46}. Finally, submesoscale motions are clearly not in geostrophic balance; they are in gradient wind balance, which means that recovering their surface velocities from only SSHA requires solving the nonlinear momentum equations (see Methods).” [Lines 220-228]

Referee #2 (Remarks to the Author):

This work presents an outstanding demonstration of the technical capabilities of the SWOT satellite to tackle open questions in physical oceanography on the role of small-scale processes in the climate system. For the very first time, SWOT enables us to

observe small structures globally and quantify the amplitude of solitary internal tides and ubiquitous small-scale eddies. This greatly advances our knowledge of small-scale ocean dynamics and helps parametrize climate models in view of more accurate climate projections.

With clear references to the building blocks of this work, the authors provide robust and reproducible conclusions.

Thank you for your comments that have improved the writing and clarified the details.

The approach and methodology are valid, and data from SWOT show unprecedented results. Comments on the data products used:

- Figures 2 and 3 are produced with the L3 ssha_unedited from the V1.0 data product. This SSHA is not filtered or edited. Averaging over 14 cycles may be enough to remove the KaRIn noise field, but there could still be residuals. Is this accounted for? Does the residual noise impact the results of Figures 2 and 3? Can the authors please comment on this?

There was a typo. Figures 1 and 3 use L2 unsmoothed, and Figures 2 and 4 use L3. We have included a more detailed description in the Methods of exactly what we do with the data before analysis to comment on this directly:

*“**SWOT data.** This study uses data from 14 complete cycles of the SWOT science orbit, from 26 July 2023 to 8 May 2024. One complete cycle comprises 584 passes and takes 21-days to complete. We use two SWOT data products: (1) ‘Level 2 Low-Rate Sea Surface Height Product – Unsmoothed’, on a 250-m grid, version PIC0. The L2 product baseline C was produced by NASA and CNES (hereafter L2); and (2) ‘Level 3 sea surface height expert’, on a 2-km grid, version 1.0, produced by the DUACS and SWOT Science Teams (hereafter L3)⁵¹.*

The L2 variable ‘ssh_karin_2’ was used in Figures 1 and 3, which is the fully corrected SSH from KaRIn relative to the reference ellipsoid, using model-based wet troposphere correction (variables ‘model_wet_tropo_cor’ and ‘sea_state_bias_cor_2’), as per recommendation from the SWOT project. We remove the mean sea surface (variable provided: ‘mean_sea_surface_cnescs’, or ‘CNES_CLS_2022’). We remove an along-track linear trend at each cross-track pixel; this removes the roll error but also some large mesoscale signal – but for our purpose of studying small-scale ocean dynamics in Figures 1 and 3 this is not an issue. We do not apply quality control flags because we want to show all the data. At certain times and locations, this unsmoothed 250-m

gridded dataset contains long swell waves (e.g., Figure 1), which are filtered out in the 2-km product.

The L3 'ssha_unedited' was used in Figures 2 and 4, which is the L2 smoothed SSHA 2-km product, but with the L3 empirical calibration replacing the L2 calibration. This is important because L3 has been reconciled with the DUACS SSHA product to allow direct comparison and has shown to be an improvement⁵². It should be noted that while the empirical calibration might absorb a fraction of the difference between SWOT and DUACS, it will be small compared to other factors (e.g., L4 smoothing)^{51,53}. The SSH anomaly is from the mean sea surface. We add back in the stationary internal tide SSH signal ('internal_tide_hret') that was removed as part of processing. We apply the variable 'quality_flag', which is used to identify and remove bad data, when computing the global variability (Figure 2; all data points with a quality flag $\neq 0$ are removed). We do not apply the flag for the study of the submesoscale eddy (Figure 4) because it removes the core of the eddy as an outlier. Because SWOT coverage and accuracy is beyond the capability of the existing mean sea surface (MSS) product, there are residual small-scale bathymetric features in the SSHA data⁵⁴. To correct for this, we compute a time mean for all cycles ($N=14$), high-pass filter it with a 50-km 2D Gaussian kernel and subtract it from each pass. This process removes the small-scale bathymetric features from the SSHA while mostly bypassing the large-scale and mesoscale time mean. This is sufficient for our purposes, but it may remove a fraction of ocean time mean signal; in the long run, a more accurate MSS product will be produced using the SWOT data⁵⁴." [Lines 424-459]

In Methods, under 'Quantifying SWOT SSH...', we discuss the residual noise contributions:

"This new view of small-scale ocean variability (Figure 2) contains within it a cornucopia of information that will take years to precisely unpack, but we know includes small-scale ocean currents and internal gravity waves (Figures 3 and 4) and SWOT residual errors³² – comprising residual instrument error including noise (e.g., there are some noticeable track errors in the high latitudes), and residual geophysical signals not fully corrected for, including the wet troposphere effect, for which scales below ~30 km are not well accounted for in model or radiometer measurements." [Lines 474-480]

- The L2 product is referred to as "karin_ssh_2" in the SWOT data. Is it the "ssh_karin_2"? If so, please correct.

Corrected.

Suggested improvements:

- Generally, SWOT's impressive technical capability and interest are clear from the examples shown. However, some physical interpretation is missing to demonstrate SWOT's potential to the community fully. For instance, in the two quantitative examples shown:

1) Why is it important to gain this knowledge on the vertically integrated time-averaged linear energy flux of solitary waves? What do these values tell us about the dynamics in the region?

We include new writing to address these important motivating questions:

"...but the fundamental question for ocean energetics can only be answered by measuring their amplitude³⁹: what is the amount and variability of their energy content and fluxes? This is essential for understanding the ocean's energy budget, but was not possible to answer, until SWOT." [Lines 158-161]

The details of the regional dynamics are out of the scope of this short article, which tries to shine a light on the bigger questions now possible to answer with SWOT data.

2) What does this kind of small eddy represent in this region? What is the benefit of observing them with SWOT? Are they always observable? What would be the long-term physical consequences of following the evolution of such structures with SWOT?

These are essential questions for us as oceanographers, but the scope of this paper limits us delving into the details. We have included several comments that address these motivating questions more broadly:

"Analysis of SWOT SSH in other high kinetic energy regions confirms the ubiquitous presence of turbulent submesoscale eddies, coherent and persistent, interacting with energetic mesoscale eddies. These submesoscale eddies can have diameters less than 10 km with Rossby numbers larger than 2 (see Extended Data Fig. 1–2, in the Kuroshio Extension). The submesoscale eddies have RMS SSHA values about 3 times larger than from the highest-resolution global ocean simulations (Extended Data Fig. 3): these SWOT observations suggest that submesoscale turbulence's contribution to the Earth's climate system is much more important than previously anticipated." [Lines 189-196]

...and...

"One hypothesis to be tested is that the newly revealed energetic small-scale eddies are likely a product of the turbulent field of strongly interacting eddies covering a large range of scales, from 1 km to 500 km^{42,47}." [Lines 241-243]

- Line 101: You mention that swells were not expected to be observed, but part of the surface long swell wave signals pass through the filter due to the high signal-to-noise ratio of KaRIn. Is this a specificity of this region? Are there regions where the signal-to-noise is not high enough? What can we learn from the knowledge of the swell? Why is it important?

It's observed globally but only for very large swell waves of severe storms; often the swell waves are not present. We have included a reference to give context to the swell waves in SWOT data:

“SWOT can resolve beyond even submesoscale processes: storm-generated surface swell waves with wavelengths of several hundred meters are revealed in the data (Figure 1). Swells were not expected to be observed because the onboard processing algorithm was designed to filter them out. However, due to the high signal-to-noise ratio of KaRIn, the long swell wave signals pass through the filter, showing up with heavily suppressed amplitudes³⁴.” [Lines 116-120]

- Line 109: Can you extend the explanation on why the pattern of the RMS differences in Fig 2 indicates that the variability is related more to ocean physics than measurement noise or errors? Is this the case also in less energetic regions?

Yes, we have extended the explanation to clarify in the Main Text...

“Significant differences appear in high kinetic energy regions such as western boundary currents (yellow boxes; Figure 2), including the Gulf Stream and Kuroshio Extension, and the Antarctic Circumpolar Current (green box). Large differences are also observed at internal gravity wave generation sites—particularly the Amazon River mouth, Mascarene Plateau, South China Sea, Andaman Sea, and Indonesian Archipelago (orange boxes)—as well as in tropical oceans and coastal zones. This collocation of the largest differences with ocean dynamical regimes indicates they are related more to ocean physics than measurement noise or errors (see Methods).” [Lines 127-134]

...and in the Methods:

“This new view of small-scale ocean variability (Figure 2) contains within it a cornucopia of information that will take years to precisely unpack, but we know includes small-scale ocean currents and internal gravity waves (Figures 3 and 4) and SWOT residual errors³² – comprising residual instrument error including noise (e.g., there are some noticeable track errors in the high latitudes), and residual geophysical signals not fully corrected for, including the wet troposphere effect, for which scales below ~30 km are not well accounted for in model or radiometer measurements.” [Lines 474-480]

- Fig2: The titles of the two panels mention SSH. Is it SSH or SSHA?

SSHA; we have now corrected this thank you.

- Fig2b shows the SSH variability for scales smaller than 50 km. What is SWOT observability in this case? Down to which scales do the authors interpret the results?

This is an active area of research, which we emphasize in Methods:

“We emphasize, however, that the first assumption is certainly not true for all SWOT-observed eddies, due in large part to the entanglement of wave signals in the SSH at scales <100 km, but also instrument measurement errors (e.g., wet troposphere effect, sea state bias, etc.) The relative contribution of these factors is variable in space and time and is an area of active research...” [Lines 539-543]

...and we also refer to Chelton’s paper (see below).

Chelton (2024) shows a Postlaunch Update on the Effects of Instrumental Measurement Errors on SWOT Estimates of Sea Surface Height, Velocity, and Vorticity (10.1175/JTECH-D-24-0035.1). How does this compare to the results in the regions investigated here?

We now include the following reference to Chelton (2024):

“Chelton (2024)⁵⁷ used postlaunch noise estimates to show resolvable feature diameters are 8.5 km and larger in velocity. Our case study has a diameter of 30 km.” [Lines 537-539]

- Line 146: is the variability of the energy and energy fluxes investigated in this work? Do you refer to spatial or temporal variability?

In writing, we highlight SWOT’s potential to observe time and space variability of the energy content and fluxes, but the example we give is a depth-integrated energy content and flux for a single wave period. We clarify this in the Methods:

“Here we compute the energy flux of the ISW packet in the Andaman Sea observed by SWOT and shown in Figure 3b, c.... To estimate a time average over one wave period, for this solitary wave packet of 5 waves generated every 12.4 hours, the magnitude is reduced...” [Lines 579-580 and 588-590]

- Line 149: have these magnitudes been observed with other models or data? Validating these fluxes is fundamental to correctly interpreting the small-scale SWOT signals. I understand that the validation of the structures shown is not the scope of this work, but the authors should comment on the importance of such a process.

We have compared with the energy flux from linear M2 internal tides in this region – it is shown globally in Figure 7 of Zhao et al. (2016) (and we got the more precise regional value of 0.8 from Zhao directly through personal communication) – and found these nonlinear solitary wave packets are much higher. For internal

solitary waves, we are not aware of any observational studies of energy flux for the Andaman Sea. However, there are some modelling studies that show the same magnitudes as we find here, and we have now included a comment on this in the manuscript:

“The linear energy flux is more than twice as large as the previously estimated coherent M2 tides⁴⁰ from altimetry, which were approximately 0.8 kW/m. The magnitudes of the total fluxes, ranging between 2-10 kW/m from SWOT, are generally consistent with existing simulations in the Andaman Sea^{59,60}.” [Lines 594-597]

- Fig 4: the image refers to cyclogeostrophy, which is explained later in the methods but not mentioned in the main text. Around lines 161 to 168, I suggest you mention that you are describing the cyclogeostrophic balance, referencing the methods section.

Thank you, we have now done this, see reply to Referee #1 above.

- Internal wave energetics (lines 317 and 319): please check if you want to refer to Figure 3 instead of Figure 4.

Corrected, thank you.

- Opportunity and challenges of SWOT:

The KaRIn noise is indeed low, but even after de-noising, some residual KaRIn noise may still be present on the swaths. This may not affect the SSH much, but accessing higher-order eddy diagnostics (velocity, vorticity) could still be challenging, and the interpretation of such quantities at small scales should be done with caution. I suggest including a comment on this as part of the challenges.

Thank you, we have now done this, see reply to Referee #1 above.

Referee #3 (Remarks to the Author):

This paper reviews the capabilities of the recently launched (Surface Water and Ocean Topography) SWOT satellite in observing fine-scale ocean dynamics, which will significantly advance our understanding of submesoscale ocean processes in ways that conventional altimetry could not. The ocean processes of scales 1-100km play crucial roles in vertical transport and energy transfer, impacting broader ocean circulation, biodiversity, and climate. Here, the authors present some of the first wide-swath satellite altimetry observations and exploit them to provide a first global statistical view of the variability at those scales. They also demonstrate how SWOT data can be used to analyse the dynamics and energy of submesoscale eddies or internal solitary waves, unveiling significant energy fluxes and dynamics often underestimated by simulations.

The first part of the paper is very valuable, as it presents how wide-swath altimetry is revolutionizing ocean topography observations, particularly by unveiling fine-scale variability that conventional altimetry systems like DUACS cannot. Unsmoothed SWOT swaths can detect much finer structures, such as filaments, internal waves, and ocean surface swell waves. By comparing SWOT data to DUACS, the authors reveal significant differences in the observed variability, especially in regions with high kinetic energy or where internal gravity waves are generated. The second part details two case studies that illustrate how SWOT data can be used to calculate the dynamics and energy of nonlinear solitary waves and submesoscale eddies. In the case of internal solitary waves, the total energy flux is stronger than coherent M2 internal tides in this region. In the case of submesoscale eddies, SWOT confirms that numerical simulations underestimate their vorticity and associated vertical velocities, minimizing their impact on the ocean dynamics and heat budget. While the methodology for the internal waves is largely based on Qiu et al. (2024), this paper offers a more accessible and comprehensive explanation of SWOT's application, easier for a broader audience to understand and use.

Given the importance of ocean submesoscale processes across multiple fields, I am confident that these examples will be widely used by researchers in various domains. It will simplify the understanding of the significant advances provided by SWOT, enabling those new to SWOT data or unfamiliar with its development to grasp its potential and begin exploring basic applications. With that in mind, the paper could benefit from some instructions, including on the necessary precautions needed to handle this data. I include some suggestions into the general comments.

The paper is very well written. The abstract is appropriate, and the introduction is valuable in highlighting the role of the submesoscale dynamics in the climate system and showing the limits of past observations of these small scale features. The presentation is sound and the figures are clear. The conclusions are valid. SWOT heralds a new era in observing ocean variability at submesoscale, and this paper clearly supports this point.

Thank you for your comments and questions that improved the details and narrative of this manuscript.

General Comments

The challenge of applying (quasi)geostrophic theories directly to SWOT SSH snapshot is mentioned at the end of the paper, but not developed. How does that impact the submesoscale eddy example? Is it fixed by applying the cyclogeostrophic balance? Which precautions should one take before applying the same method on another

submesoscale eddy? The paper would benefit from further instructions on when similar computation could be applied or not.

Thank you, we have now done this, see reply to Referee #1 above.

Similarly, how is the SSH noise impacting the test cases? Chelton et al. (2022) anticipated that additional smoothing would be required for most applications of SWOT estimates of velocity and vorticity. Has any of this additional smoothing been applied here, and when should it be considered when reproducing this analysis on other submesoscale eddies?

We have now given specific details in the Methods:

“When we compute velocity, we smooth the L3 SSHA field with a 2D boxcar filter of 6 km (3 grid points) before taking the horizontal gradients. This suppresses any residual noise due to high surface waves.” [Lines 535-537]

...and...

“We emphasize, however, that the first assumption is certainly not true for all SWOT-observed eddies, due in large part to the entanglement of wave signals in the SSH at scales <100 km, but also instrument measurement errors (e.g., wet troposphere effect, sea state bias, etc.) The relative contribution of these factors is variable in space and time and is an area of active research...” [Lines 539-543]

Chelton et al. (2022)’s findings were updated by Chelton (2024) after launch with the actual noise levels, which we discuss above in the reply to Referee #2.

There is no mention of measurement, geophysical correction, or mapping uncertainty in the paper. Similarly, the Sea Surface Height 1d plots in Figures 3c and 4e show no error bars or confidence intervals. Having a broad idea of the order of magnitude of this uncertainty would be valuable to the reader (How big is it? Is it the same within the whole swath?)

We’ve now added:

“For the two case studies shown here (Figures 3 and 4), the signal strength of each feature is $O(10\text{ cm})$. For context, the error standard deviation of SWOT at 6-km resolution is below 0.5 cm. So, these examples are well above the noise floor. However, caution should be taken when analyzing SWOT data globally, since both signal and errors are time and space dependent.” [Lines 482-485]

We have also included a discussion of sources of error in the Methods (see above).

Other comments:

L 89: “new small-scale features”. Not really new but detectable for the first time.

Agreed, we have removed the word ‘new’.

L94: The figure does not clearly show the “smaller-scale eddies” mentioned here.

Agreed, we have reworded to:

“On the same day, instantaneous SWOT observations unveil the presence of not only the mesoscale eddy but a cornucopia of submesoscale features, including waves radiating eastward away from the Plateau.” [Lines 108-110]

L 96: The wavelength is 30-km in the figure.

Thank you, we have corrected the text and incorporated the variability of wavelengths seen in the image:

“The waves are nonlinear internal solitary wave packets with wavelengths from several to tens of kilometers and SSHA amplitudes up to 30 cm.” [Lines 110-112]

L 101-106: The conclusion from is a bit unclear to me. While it shows the impressive capability of SWOT in detecting small scale signals of only hundreds of meters, it also mentions that this signal was supposed to be filtered out. Does that imply that the user should filter himself this signal when looking at larger scale (1-100km) processes? Is it totally corrected in the L3 dataset?

We have included a short explanation in Methods:

“At certain times and locations, this unsmoothed 250-m gridded dataset contains long swell waves (e.g., Figure 1), which are filtered out in the 2-km product.” [Lines 438-440]

L112: RMS or STD?

For SSHA, RMS is the same as STD. But for clarity, we have removed all use of the term STD now in the paper.

L 115-116: “20% of the total SSHA RMS” Where does this percentage come from? This needs to be complemented with a map of the total SSHA RMS or a paper showing one should be added here.

We have removed this sentence during the revisions.

L129: “30% of SSHA variability”

We have removed this sentence during the revisions.

L145-146: “but the fundamental question for ocean energetics can only be answered by measuring their amplitude”.

Corrected.

L 158: “warmer”. It looks colder from the Figure.

Corrected.

L 168-170: Reference?

We have added a reference to *Klein et al. (2019)*.

L174-176 and Extended Data Fig. 3. I am not sure the high-resolution model has been subsampled on SWOT observations. Does the subsampling impact the RMS SSH values? It seems to me that the comparison would make more sense if the SWOT values were compared with the direct high-resolution model outputs.

We show the Ilc4320 model dataset interpolated onto the SWOT swath, available publicly:

https://podaac.jpl.nasa.gov/dataset/SWOT_SIMULATED_L2_KARIN_SSH_ECCO_LLC_4320_SCIENCE_V1

We use the variable ‘*simulated_true_ssh_karin*’ that is the true model SSH on the SWOT swath grid. This does not have a significant impact on the RMS SSH values.

L185-186: Refer to Figure

Reference added.

Extended Data 2: Arrows?

We have removed the arrows in Extended Data Figure 2.

References 27 and 41 might have titles swapped. I was not able to access them.

Corrected, thank you.

L251: What is the grid resolution of the L3 dataset?

2-km. We have included this in Methods.

L267: It would be nice to have the time period over which this standard deviation has been computed.

We had included it under 'SWOT Data' in the Methods. We've now included it in this sentence as well for clarity.

L317: I think you're referring to Fig. 3.

Corrected, thank you.

L319: same as the previous comment

Corrected, thank you.